# SUBJECTIVE LEARNING FOR OPEN-ENDED DATA

## ABSTRACT

Conventional supervised learning typically assumes that the learning task can be solved by learning a *single* function since the data is sampled from a fixed distribution. However, this assumption is invalid in open-ended environments where no task-level data partitioning is available. In this paper, we present a novel supervised learning framework of learning from *open-ended data*, which is modeled as data implicitly sampled from multiple domains with the data in each domain obeying a domain-specific target function. Since different domains may possess distinct target functions, open-ended data inherently requires *multiple* functions to capture all its input-output relations, rendering training a single global model problematic. To address this issue, we devise an Open-ended Supervised Learning (OSL) framework, of which the key component is a *subjective function* that allocates the data among multiple candidate models to resolve the "conflict" between the data from different domains, exhibiting a natural hierarchy. We theoretically analyze the learnability and the generalization error of OSL, and empirically validate its efficacy in both open-ended regression and classification tasks.

## 1 INTRODUCTION

A hallmark of general intelligence is the ability of handling open-ended environments, which roughly means complex, diverse environments with no manual task specification (Adams et al., 2012; Clune, 2019; Colas et al., 2019; Wang et al., 2020). Conventional supervised learning typically assumes that the learning task can be solved by approximating a single ground-truth target function (Vapnik, 2013). However, this assumption is invalid in open-ended environments where the data may implicitly belong to multiple, disparate *domains* with potentially different target functions since no manual task-level data partitioning is available. For instance, when collecting image-label pairs from the Internet, an image of a red sphere can correlate with the label of both "red" and "sphere", implicitly representing two distinct domains or "metaconcepts" (Han et al., 2019): "color" and "shape". While being generic, this setting also characterizes many practical scenarios where the data comes from multiple sources, contexts or groups: for example, in federated learning the data is distributed on multiple clients, and in algorithmic fairness research the data is from different populations. In these scenarios, the input-conditional label distribution may vary in different domains (also referred to as *concept shift* (Kairouz et al., 2021)) due to personal preferences or other latent factors, corresponding to different target functions.

Since different domains may possess different target functions, training a single global model by running Empirical Risk Minimization (ERM) using all data is problematic due to the potential "conflict" between the data from different domains: in the "red sphere" example above, directly training on all data will lead to the unfavorable result of "50% red, 50% sphere". Similar phenomena have also been observed by prior works (Finn et al., 2019; Su et al., 2020), where a learner simultaneously regressing from multiple target functions trivially outputs their mean. This indicates that the data sampled from open-ended environments, which we refer to as *open-ended data*, exhibits a structural difference from conventional supervised data. Formally, we introduce a novel, dataset-level measure named *mapping rank*, which represents the minimal number of functions required to "fully express" all input-output relations in the data and can be used to expound such difference more clearly.

**Definition 1** (Mapping rank). *Let $\mathcal{X}$ be an input space, $\mathcal{Y}$ an output space, and $Z = \{(x_i, y_i)\}_{i=1}^{l}$ a dataset with cardinality $l$. Let $F(r) = \{f_i\}_{i=1}^{r}$ be a function set with cardinality $r$, where each element is a single-valued deterministic function from $\mathcal{X}$ to $\mathcal{Y}$. Then, the mapping rank of $Z$, denoted*

*by $R := R(Z)$, is defined as the minimal positive integer $r$, satisfying that there exists a function set $F(r)$ such that for every $(x, y) \in Z$, there exists $f \in F(r)$ with $f(x) = y$.*

Note that here we assume that there exist deterministic relations between inputs and outputs in the same domain, which is generally a mild assumption and is satisfied in many practical applications. Under Definition 1, conventional supervised data yields a mapping rank $R = 1$ as it assumes that the whole dataset can be expressed by a single function. In contrast, open-ended data has a mapping rank $R > 1$ since for the same input different outputs exist, manifested as the conflict between data samples that renders training a single model problematic. Hence, it is natural to consider allocating the data to multiple models, so that the data processed by each model has a mapping rank $R = 1$ and thus can be handled with ERM. Although in some scenarios, apart from data samples there also exists side-information or metadata that can be exploited to identify the domains, this information may be difficult to define or collect in practice (Hanna et al., 2020; Creager et al., 2021); even when such side-information is available, in many cases it still remains unclear how to leverage such information to detect and resolve the potential conflict between domains. Therefore, the problem of how to properly allocate open-ended data among the models is highly non-trivial.

To tackle the aforementioned challenge, we present an *Open-ended Supervised Learning (OSL)* framework to enable effective learning from open-ended data. Concretely, OSL maintains a set of low-level models and a high-level *subjective function* that automatically allocates the data among these models so that the data processed by each model exhibits no conflict. The term "subjective" is used since in conventional supervised learning such allocation is manually performed during the data collecting process and thus conforms to human subjectivity. The motivation of such process is that if the subjective function yields an inappropriate allocation, i.e., assigning conflicting data samples to the same model, then it will hinder the global minimization of the training error due to the conflict, which in turn drives the subjective function to alter its allocation strategy. Using a probablistic reformulation of OSL, we establish the connection between the data allocation and the posterior maximization using a variant of Expectation-Maximization (EM) algorithm (Dempster et al., 1977), and show that the optimal form of the subjective function can be explicitly derived.

Theoretically, we respectively analyse the Probably Approximately Correct (PAC) learnability (Valiant, 1984) and the generalization error of OSL. Using the tools from statistical learning theory (Vapnik, 2013), we show that the relation between the number of low-level models and the mapping rank of data plays a key role in the learnability of OSL, and the generalization error of OSL can be decomposed into terms that respectively reflect high-level data allocation and low-level prediction errors. Empirically, we conduct extensive experiments including simulated open-ended regression and classification tasks to verify the efficacy of OSL. Our results show that OSL can effectively allocate and learn from open-ended data without additional human intervention. In summary, our contributions are three-fold:

**Open-ended data and mapping rank.** We formalize a new problem of learning from open-ended data, and introduce a novel measure termed as mapping rank to outline the structural difference between open-ended data and conventional supervised data.

**OSL framework with theoretical guarantee.** We present an OSL framework to enable effective learning from open-ended data (Section 2), and theoretically justify its learnability (Section 3.1) and generalizability (Section 3.2) respectively.

**Empirical validation of efficacy.** We conduct extensive experiments on both open-ended regression and classification tasks. Experimental results validate our theoretical claims and demonstrate the efficacy of OSL (Section 4).

## 2 Open-Ended Supervised Learning

In this section, we present the overall formulation and the algorithm of OSL. We adhere to the conventional terminology in supervised learning, and let $\mathcal{X}$ be an input space, $\mathcal{Y}$ an output space, $\mathcal{H}$ a hypothesis space where each hypothesis (model) is a function from $\mathcal{X}$ to $\mathcal{Y}$, and $\ell : \mathcal{Y} \times \mathcal{Y} \to [0, 1]$ a non-negative and bounded loss function without loss of generality. We use $[k] = \{1, 2, \cdots, k\}$ for positive integers $k$, and denote by $\mathbb{1}(\cdot)$ the indicator function. We use superscripts to denote sampling indices (e.g., $d^i$ and $x^{ij}$) and subscripts as element indices (e.g., $d_i$).

## 2.1 PROBLEM STATEMENT

We begin by introducing the notion of *domain* to formulate the generation process of open-ended data. Inspired by Ben-David et al. (2010), we define a domain $d$ as a pair $\langle P, c \rangle$ consisting of a distribution $P$ on $\mathcal{X}$ and a deterministic target function $c : \mathcal{X} \to \mathcal{Y}$, and assume that the open-ended data is generated by a *domain set* $D = \{d_i\}_{i=1}^N = \{\langle P_i, c_i \rangle\}_{i=1}^N$ containing $N$ (agnostic to the learner) domains. Each domain has its own sub-dataset $Z_i = \{(x_{ij}, y_{ij})\}_{j=1}^{l_i}$ with cardinality $l_i$, where $x_{i1}, \cdots, x_{il_i}$ are i.i.d. drawn from $P_i$ and $y_{ij} = c_i(x_{ij})$. The whole dataset is the union of these sub-datasets: $Z = \bigcup_{i=1}^N Z_i$ with cardinality $l = \sum_{i=1}^N l_i$ and mapping rank $1 < R \le N$. We then consider a bilevel sampling procedure: first, $m$ domain samples $d^1, \cdots, d^m$ are i.i.d. drawn from a distribution $Q$ defined on $D$ (the same domain may be sampled multiple times), resulting in $m$ sampling *episodes*; second, in each sampling episode $n$ data samples are i.i.d. drawn from the sub-dataset corresponding to the sampled domain. Hence, the dataset or any sub-datasets may be sampled multiple times during training. This sampling regime is analogous to the bilevel sampling process adopted by meta-learning. However, meta-learning usually assumes a dense distribution of related domains to enable task-level generalization (Pentina & Lampert, 2014; Amit & Meir, 2018), while OSL is compatible with scarce and disparate domains and inter-domain transfer is orthogonal.

As we have mentioned in Section 1, a single model is not sufficient in this setting when $R > 1$. Thus, we equip the learner with a *hypothesis set* $H = \{h_i\}_{i=1}^K$ consisting of $K > 1$ hypotheses, enhancing its expressive capability. Although both $N$ and $K$ are assumed to be unknown, we will show that in general $K \ge R$ suffices (Section 3.1), which eases the difficulty of setting the hyperparameter $K$.

In the above setting, an episodic sample number parameter $n$ is introduced to maintain the local consistency of data, implicitly assuming that we are able to sample a size-$n$ data batch at a time from each domain. While this formulation subsumes the fully online case of $n = 1$, we note that although sometimes setting $n = 1$ works in practice, it also tends to be risky since it may raise difficulties in controlling the generalization error, as we will both theoretically and empirically demonstrate in the following sections (see Section 3.2 and Section 4.3).

## 2.2 GLOBAL ERROR

In this section, we present the global learning objective of OSL. Since the open-ended dataset implicitly contains multiple underlying input-output mapping relations, a primary start point can be the empirical multi-task loss with pre-defined data-domain correspondences:

$$\widehat{er}_{\text{MTL}}(H) = \frac{1}{m} \sum_{i=1}^m \frac{1}{n} \sum_{j=1}^n \ell \left[ h_{\text{ORACLE}(i)} \left( x^{ij} \right), y^{ij} \right], \tag{1}$$

where $\text{ORACLE} : [m] \to [K]$ is an oracle mapping function that determines which hypothesis each data batch $Z^i := \{(x^{ij}, y^{ij})\}_{j=1}^n$ in episode $i$ is assigned to. However, in OSL the oracle mapping function is unavailable, imposing a fundamental discrepancy. To tackle this difficulty, here we substitute the oracle mapping function with a learnable *empirical subjective function* $\hat{g} : \mathcal{H}^K \times \mathcal{X}^n \times \mathcal{Y}^n \to H$ that aims to select a hypothesis $h$ from the hypothesis set $H$ for the data batch $Z^i$. This substitution yields the empirical global error of OSL:

$$\widehat{er}(H) := \frac{1}{m} \sum_{i=1}^m \frac{1}{n} \sum_{j=1}^n \ell \left[ \hat{g} \left( H, Z^i \right) \left( x^{ij} \right), y^{ij} \right]. \tag{2}$$

Our insight is that the data batch itself can be harnessed to guide its suitable allocation in the presence of mapping conflict: intuitively, a single model trained by conflicting data batches result in an inevitable training error, thus hindering the minimization of the global error. This in turn facilitates data allocation with less conflict. Given the empirical error 2, the corresponding expected global error is

$$er(H) := \mathbb{E}_{d_i \sim Q} \mathbb{E}_{x \sim P_i} \ell \left[ g(H, d_i)(x), c_i(x) \right], \tag{3}$$

where $g : \mathcal{H}^K \times D \to H$ is an *expected subjective function*, which can be viewed as the empirical subjective function with infinite samples from every single domain so that all available domain information can be fully reflected by the data samples. The global objective 3 characterizes the scenario where the learner interacts with only one domain in a particular time period (which is natural in the real world), therefore its performance can be separately tested in all domains.

So far, our framework remains incomplete since the subjective function remain undefined. In the next section, we will present our design of the subjective function and elucidate its rationale.

## 2.3 Derivation of Subjective Function

To attain a reasonable choice of the subjective function, in this section we provide an alternative, probabilistic view of OSL from the angle of maximum conditional likelihood, and draw an intriguing connection between the choice of the subjective function and the posterior maximization using a variant of the EM algorithm (Dempster et al., 1977) on open-ended data. Concretely, let $p(Y \mid X)$ represents the predictive conditional distribution of the hypothesis set $H$, where we use $X$ and $Y$ as the shorthand for $(x^{11}, x^{12}, \cdots, x^{mn})$ and $(y^{11}, y^{12}, \cdots, y^{mn})$ respectively. We consider maximizing the empirical log-likelihood $\log p(Y \mid X) = \sum_{i=1}^{m} \sum_{j=1}^{n} \log \sum_{k=1}^{K} p(y^{ij}, h = h_k \mid x^{ij})$ using EM, where $h$ denotes the selected hypothesis. Accordingly, in the $i$-th sampling episode, in the E-step we aim to estimate the model posterior $P(h = h_k \mid Z^i)$ that represents the responsibility of the $k$-th hypothesis in the hypothesis set w.r.t. the local data batch $Z^i = \{(x^{ij}, y^{ij})\}_{j=1}^{n}$, while in the M-step we seek to maximize

$$\mathcal{L}(H) = \sum_{k=1}^{K} P\left(h = h_k \mid Z^i\right) \sum_{j=1}^{n} \log \left[\pi_k p\left(y^{ij} \mid x^{ij}, h_k\right)\right], \tag{4}$$

where $\pi_k := P(h = h_k) > 0$ denotes the prior of the $k$-th hypothesis in $H$. This draws a direct connection between OSL and EM: the E-step corresponds to the functionality of the subjective function that chooses a hypothesis for a given data batch; the M-step corresponds to updating the selected hypothesis by minimizing the empirical prediction error in the episode. The main difference is that here we assume the subjective function to be *deterministic*, representing a "hard" assignment of the data to the model. This entails the usage of a variant known as hard EM (Samdani et al., 2012), which considers the posterior to be a Dirac delta function. Applying this constraint, the E-step of EM yields $h = \arg\max_{h_k \in H} \sum_{j=1}^{n} \log p(y^{ij} \mid x^{ij}, h_k)$ under a uniform prior, motivating a principled choice of the subjective function:

$$\hat{g}\left(H, Z^i\right) = \arg\min_{h \in H} \sum_{j=1}^{n} \ell\left[h(x^{ij}), y^{ij}\right], i \in [m], \tag{5a}$$

$$g(H, d_i) = \arg\min_{h \in H} \mathbb{E}_{x \sim P_i} \ell\left[h(x), y\right], i \in [N], \tag{5b}$$

which can be interpreted as selecting the hypothesis that incurs the smallest (empirical or expected) error. While this connection is not rigorous in general, in some cases exact equivalence can be obtained when certain types of loss functions and likelihood families are applied, which encompasses common regression and classification settings. We provide concrete analysis in Appendix A.

## 2.4 Overall Algorithm

In practice, we assume that the hypothesis set $H$ comprises $K$ parameterized hypotheses with parameter vectors $\Theta = (\theta_1, \theta_2, \cdots, \theta_K)$ respectively. In the high level, with the choice of the empirical subjective function 5a, our algorithm consists of two phases in each sampling episode: (i) evaluating the error of each hypothesis in $H$ w.r.t. the data in this episode, and (ii) training the hypothesis with the smallest error. For brevity, we introduce a notion of *empirical episodic error* defined as

$$\widehat{er}^i(h; \theta) := \frac{1}{n} \sum_{j=1}^{n} \ell\left[h\left(x^{ij}; \theta^i\right), y^{ij}\right], i \in [m], \tag{6}$$

where $h \in \mathcal{H}$ is a hypothesis parameterized by $\theta$. Then, phase (i) aims to find a hypothesis that mininize 6. Note that this selection process may induce a bias between empirical and expected objectives 2 and 3, since a hypothesis that minimizes the empirical loss on finite samples may not minimize the expected loss of this domain. Hence, the global error of OSL can be intuitively decomposed into a high-level subjective error that measures the reliability of the model selection, and a low-level model error that measures the accuracy of models, of which we provide detailed theoretical analysis in Section 3.2. In practice, we parameterize each hypothesis in the hypothesis set with a deep neural network (DNN), and apply stochastic gradient descent (SGD) for the optimization process. We provide the pseudo-code of OSL in Appendix B.

# 3 THEORETICAL ANALYSIS

In this section, we present the theoretical analysis on OSL. All proofs are deferred to Appendix C.

## 3.1 LEARNABILITY

We first analyze the learnability of OSL based on PAC learnability (Valiant, 1984). Since our analysis directly applys to conventional supervised learning by setting $K = 1$, we also verify the conflict phenomenon mentioned in Section 1 from a theoretical perspective. While the learnability in conventional PAC analysis mainly relates to the choice of the hypothesis space, open-ended data imposes a new source of complexity by its mapping rank, and we expect the cardinality of the proposed hypothesis set can compensate this complexity. We consider the realizable case where the hypothesis space covers the target functions in all domains, which helps to underline the core characteristic of our problem. We begin by a result on the form of the optimal solutions of OSL.

**Proposition 1** (Form of the optimal solutions). *Assume that the target functions in all domains are realizable. Then, the following two propositions are equivalent:*

(1) *For all domain distributions $Q$ and data distributions $P_1, P_2, \cdots, P_N$, $er(H) = 0$.*

(2) *For each domain $d = \langle P, c \rangle$ in $D$, there exists $h \in H$ such that $\mathbb{E}_{x \sim P}\ell[h(x), c(x)] = 0$.*

Proposition 1 suggests that minimizing the expected global error 3 with 5b elicits a global optimal solution where every target function is learned accurately in the realizable case. Note that this does not require $N$ hypotheses for $N$ domains, since non-conflict domains can be incorporated into the same model. In other words, what determines the minimal cardinality of the hypothesis set is not the number of the domains, but the number of *conflicting domains*, which can be exactly characterized by the mapping rank. Formally, we attain a necessary condition of the PAC learnability of OSL.

**Theorem 1** (Learnability). *A necessary condition of the PAC learnability of OSL is $K \geq R$.*

Theorem 1 indicates that the cardinality of the hypothesis set should be large enough to enable effective learning, and shows the impact of mapping rank on the learnability of OSL.

**Remark 1.** While it is generally hard to derive a necessary and sufficient condition of PAC learnability theoretically (which requires a sample-efficient optimization algorithm), we empirically find that $K \geq R$ is indeed an essential condition for learnability with complex hypothesis spaces (parameterized DNNs). We also note that several recent works (Allen-Zhu et al., 2019; Du et al., 2019) have proved that over-parameterized neural networks trained by SGD can achieve zero training error in polynomial time under non-convexity, which may also be used to enhance our analysis. We leave a more rigorous study for future work.

## 3.2 GENERALIZATION ERROR

We have shown that minimizing the expected global error is sufficient for effective learning from open-ended data. However, in practice, since we only have access to the empirical global error, how to control the discrepancy between these two errors, i.e., the generalization error, remains crucial. In this section, we identify the terms in the generalization error that respectively correspond to the high-level subjective error and the low-level model error of OSL, and discuss their controlling strategies. The key results are (i) the number of episodes and episodic samples can compensate each other in controlling the low-level model error, and (ii) the number of episodic samples is critical for controlling the high-level subjective error. We have the following theorem:

**Theorem 2** (Generalization error bound). *For any $\delta \in (0, 1]$, the following inequality holds uniformly for all hypothesis sets $H \in \mathcal{H}^K$ with probability at least $1 - \delta$:*

$$er(H) \leq \widehat{er}(H) + \sqrt{\frac{\mathsf{VC}(\bar{\mathcal{S}})\left(\ln 2m/\mathsf{vc}(\bar{s}) + 1\right) - \ln \delta/12}{m}} + \frac{1}{m} \tag{7a}$$

$$+ \sum_{k=1}^{N} \left( \frac{m_k}{m} \sqrt{\frac{\mathsf{VC}(\mathcal{S})\left(\ln 2m_k n/\mathsf{vc}(\mathcal{S}) + 1\right) - \ln \delta/12N}{m_k n}} + \frac{1}{mn} \right) \tag{7b}$$

$$+ 2\sqrt{\frac{\mathsf{VC}(\mathcal{S})\left(\ln 2n/\mathsf{vc}(\mathcal{S}) + 1\right) - \ln \delta/24m}{n}} + \frac{2}{n}, \tag{7c}$$

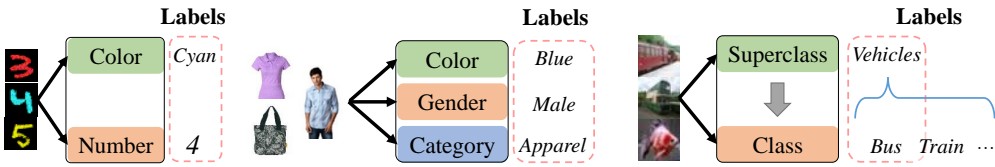

Figure 1: Open-ended classification tasks and datasets.

*where $\bar{S} := \{\langle P, c \rangle \mapsto \mathbb{E}_{x \sim P} \ell [h(x; \theta), c(x)]\}, \theta \in \Theta$ is the function set of the domain-wise expected error, $S := \{(x, y) \mapsto \ell [h(x; \theta), y]\}, \theta \in \Theta$ is the function set of the sample-wise error, $m_k := \sum_{i=1}^{m} \mathbb{1} (c^i = c_k)$ is the sampling count of the target function from the k-th domain $d_k (k \in [N])$, and $\mathsf{VC}(\cdot)$ the Vapnik-Chervonenkis (VC) dimension (Vapnik & Chervonenkis, 1971).*

Theorem 2 indicates that the expected global error is bounded by the empirical global error plus three terms. The *subjective estimation error* term 7c is derived by bounding the discrepancy between the empirical and the expected subjective functions due to the limitation of finite episodic samples (detailed derivation is in Appendix C.3). This term can be controlled by the sample-level complexity term $\mathsf{VC}(S)$ and the number of episodic samples $n$, which certifies the necessity of the local consistency assumption in Section 2. Although in theory this error term converges to zero only if $n \to \infty$, in practice we find that usually a very small $n$ (e.g., $n = 2$) suffices (see Section 4). We posit that this is because the domains in our experiments are relatively diverse, thus reducing the difficulty of discriminating between different domains. The *domain estimation error* term 7a contains a domain-level complexity term $\mathsf{VC}(\bar{S})$, and converges to zero if the number of episodes reaches infinity ($m \to \infty$); the *instance estimation error* term 7b contains sample-level complexity terms $\mathsf{VC}(S)$, and converges to zero if the sample number in each episode *or* the number of episodes reaches infinity ($n \to \infty$ *or* $m \to \infty$), showing the synergy between high-level domain samples and low-level data samples in controlling the model-wise generalization error.

**Comparison with existing bounds.** We compare our bound 7 with existing bounds of conventional supervised learning (Vapnik, 2013; McAllester, 1999) and meta-learning (Pentina & Lampert, 2014; Amit & Meir, 2018). Typically, supervised learning bounds contain a instance-level complexity term as 7b, and meta-learning bounds further contain a task-level complexity term as 7a. Yet, conventional supervised learning only considers a single domain or multiple *known* domains, while meta-learning treats each episode as a *new* domain rather than domains that may have been encountered as in OSL. Thus, none of these bounds contains an explicit inference term as 7c.

**Remark 2.** While our bound applies VC dimension as the complexity measure, extensions to other data-dependant complexity measures such as Rademacher and Gaussian complexities (Bartlett & Mendelson, 2002; Koltchinskii & Panchenko, 2000) is straightforward. It is worth noting that the bounds based on these measures share the same asymptotic property w.r.t. $m$ and $n$ as in the bound 7.

## 4 EXPERIMENTS

In this section, we report experimental results on two basic supervised learning tasks with open-ended data: regression and classification. Our experiments are designed to (i) validate our theoretical claims, (ii) assess the effectiveness of OSL, and (iii) compare OSL with task-specific baselines.

### 4.1 SETUP AND BASELINES

**Open-ended regression.** We consider an open-ended regression task in which data points are sampled from three heterogeneous functions with varied shapes, as shown in Figure 2a (solid lines). The details of this task are in Appendix D.

To the best of our knowledge, there is no off-the-shelf method that can automatically disentangle the data from different domains in the open-ended regression setting. The most relevant approach from our view is meta-learning that learns a global model which can be fastly fine-tuned to fit each domain (this is different from OSL, which assigns different models to conflicting domains explicitly). Hence, we compare OSL with three baselines. (1) *Vanilla*: a conventional ERM-based learner. (2) *MAML* (Finn et al., 2017): a popular gradient-based meta-learning approach. (3) *Modular* (Alet

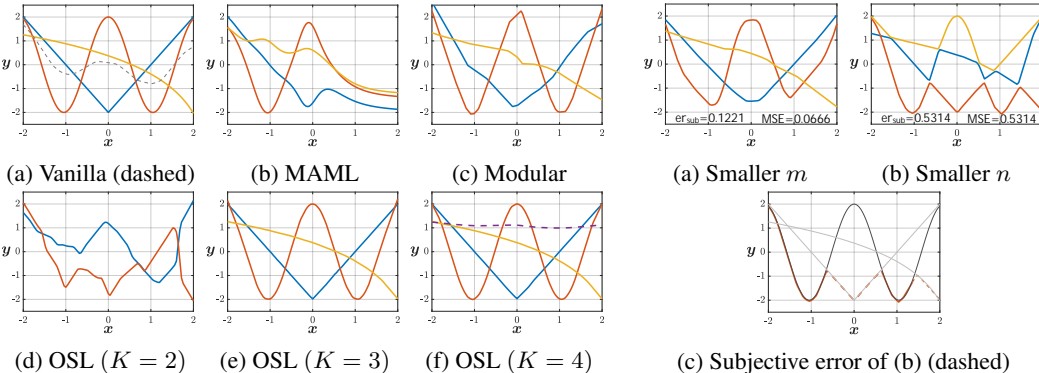

Figure 2: Results of OSL and the baselines on the open-ended regression task. (a) Ground-truth functions (solid) and the result of Vanilla (dashed). (b)(c) Results of MAML and Modular. (d)(e)(f) Results of OSL with different number of low-level models (solid). The dashed line in (f) indicates that OSL abandons a redundant model.

Figure 3: Impact of sampling hyperparameters on OSL. (a) Fewer episodes ($m = 50$). (b) Fewer episodic samples ($n = 1$). (c) The subjective error when $n = 1$ (dashed lines represent incorrect data allocations).

et al., 2018): a modular meta-learning approach that extends MAML using multiple modules. We set the hyperparameters of OSL to $K = 3, m = 250$ and $n = 2$. To verify our theoretical results, we also run OSL with different number of hypotheses ($K = 2$ and $K = 4$) and different sampling hyperparameters ($m = 50, n = 2$ and $m = 250, n = 1$). More details are in Appendix D.

In addition, we demonstrate the effectiveness of OSL on a real-world multi-dimensional regression task; details and results are in Appendix E.1.

**Open-ended classification.** We consider two types of open-ended image recognition tasks where the same image may correspond to different labels in different sample pairs. We refer to these tasks according to the structure of their label spaces, namely *parallel* and *hierarchical* tasks. For parallel tasks, we derive the data respectively from *Colored MNIST*, a variant of *MNIST* where each digit is assigned with a digit label and a color label, and *Fashion Product Images* (Aggarwal, 2019), a multi-attribute clothes dataset that involves 3 main parallel tasks including gender, category and color classification, as shown in Figure 1a; we construct open-ended datasets by randomly choosing one label from the label set for each image. For the hierarchical task, we derive the data from *CIFAR-100* (Krizhevsky & Hinton, 2009), a widely-used image recognition dataset comprising 100 classes with "fine" labels subsumed by 20 superclasses with "coarse" labels, as shown in Figure 1b; we construct the open-ended dataset by randomly using the fine or the coarse label for each image.

In classification, the most relevant problem setting to OSL is multi-label learning (Zhang & Zhou, 2014), which considers the scenario where an input $x$ is related with a label set $\boldsymbol{y} = \{y_i\}_{i=1}^N$. This setting is similar to OSL in terms that both OSL and multi-label learning considers the scenario where the same input may relate to multiple labels. However, the key difference is that multi-label learning requires that all labels in the label set are provided *simultaneously*, while in OSL each data sample only contains *one* label $y_i \in \boldsymbol{y}$. Therefore, the setting of OSL can be alternatively modeled as "multi-label learning with missing labels", i.e., in each sample $N - 1$ labels in $\boldsymbol{y}$ are missing and only one label remains (note that this is a very extreme setting). Hence, we compare OSL with the following baselines. (1) *Probabilistic concepts (ProbCon)*: this baseline directly models the relation between inputs and outputs using an conditional probability distribution (Devroye et al., 1996). We adopt a single-model DNN to learn this probability distribution using the cross-entropy loss and choose top-$N$ labels as the final prediction results. (2) *Semi-supervised multi-label learning*: this class of methods model open-ended classification as the problem of "multi-label learning with missing labels" (Yu et al., 2014; Bi & Kwok, 2014; Huang et al., 2019), i.e., only one label in each label set is available. We compare OSL with two representative semi-supervised methods: *Pseudo-label (Pseudo-L)* (Lee, 2013) and *Label propagation (LabelProp)* (Iscen et al., 2019). We also introduce two oracle baselines using additional information. (3) *Full labels (Full-L)*: a standard multi-label learning method where we provide the full label set for each image, hence there is no missing labels. (4) *Full tasks (Full-T)*: a standard multi-task learning method where the "task" of

each image is designated by human experts in advance to ensure that there is no conflict within each task. More details on baselines can be found in Appendix D.

To further demonstrate the applicability of OSL, we also conducted an experiment on *Fashion Product Images* with simulated concept shift between domains with the *same* label space; details and results are in Appendix E.2.

## 4.2 EVALUATION METRICS

Since the global error of OSL is related to the error of both the high-level subjective function and the low-level models, we respectively propose two metrics to quantitatively estimate these errors.

**Subjective error.** This metric measures the learner's ability to perform appropriate data allocation. Given a domain $d$, a suitable subjective function should yield stable allocations for all data batches sampled from this domain. Thus, we measure the error of the subjective function using the rate of *inconsistent data allocations*, which we define as

$$\text{SUBERR}(d) = 1 - \max_{h \in H} \frac{1}{l_d} \sum_{j=1}^{l_d} \mathbb{1} \left[ g \left( H, z^j \right) = h \right],\tag{8}$$

where $l_d$ denotes the total number of samples in domain $d$ (the same below).

**Model error.** This metric measures the learner's ability to make accurate in-domain predictions, which is analagous to traditional single-task error metrics. Given a domain $d$, it takes the form of

$$\text{MODERR}(d) = \min_{h \in H} \frac{1}{l_d} \sum_{j=1}^{l_d} e \left[ h \left( x_j \right), y_j \right],\tag{9}$$

where we apply $e(y, y') = (y - y')^2$ for regression and $e(y, y') = \mathbb{1}(y \neq y')$ for classification.

## 4.3 EMPIRICAL RESULTS AND ANALYSES

**Open-ended regression.** We compare the performance of OSL and the baselines in Figure 2. Unsurprisingly, the vanilla baseline converges to a trivial mean function (dashed curve in Figure 2a). MAML successfully predicts the left part of all target functions by fine-tuning from episodic samples, but fails in the right part where functions exhibit larger difference. We hypothesis that it is because meta-learning typically requires tasks to be in sufficient numbers, and with more similarity. Although Modular correctly predicts the general trend of the curves, its predictions are still inaccurate in fine-grained details. Note that both meta-learning methods use more episodic data samples than OSL (see Appendix D.3.1). Meanwhile, OSL with $K \geq R$ ($K = 3$ or $4$) successfully distinguishes different functions and recover each of them accurately while OSL with $K < R$ ($K = 2$) fails, which matches our analysis

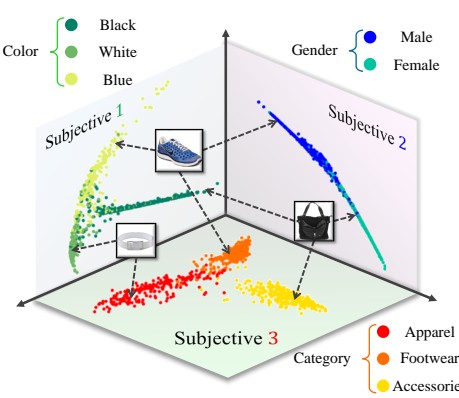

Figure 4: Visualization of the features output by OSL in *Fashion Product Images*.

in Section 3.1. In particular, OSL with $K = 4$ automatically leaves one network to be redundant (dashed curve in Figure 2f), demonstrating the robustness of our algorithm.

Figure 3 illustrates the impact of sampling hyperparameters on OSL. Concretely, OSL with fewer sampling episodes $m = 50$ induces a large model error 9, which corresponds to the sample-wise estimation term in the generalization error 7b since the product $mn$ is not sufficiently large. On the other hand, OSL with fewer episodic samples $n = 1$ induces a large subjective error 8, which corresponds to the subjective estimation term in the generalization error 7c. Another interesting phenomenon is that the curves in Figure 3b are partially swapped compared with the ground truth when $n = 1$, indicating wrong data allocation, which also corroborates our theory.

Table 1: Results of OSL and the baselines on open-ended classification tasks. We report subjective and model errors on *Number (Num)* and *Color (Col)* domains of *Colored MNIST*, *Gender (Gen)*, *Category (Cat)* and *Color (Col)* domains of *Fashion Product Images*, and *Superclass (Sup)* and *Class (Cla)* domains of *CIFAR-100* respectively.

| Methods | Colored MNIST | | | | Fashion Product Images | | | | | | CIFAR-100 | | | |
|---|---|---|---|---|---|---|---|---|---|---|---|---|---|---|
| | SUBERR | | MODERR | | SUBERR | | | MODERR | | | SUBERR | | MODERR | |
| | *Num* | *Col* | *Num* | *Col* | *Gen* | *Cat* | *Col* | *Gen* | *Cat* | *Col* | *Sup* | *Cla* | *Sup* | *Cla* |
| ProbCon | **0.09** | 0.34 | 3.04 | 0.39 | 8.81 | 13.54 | 12.50 | 22.80 | 18.02 | 33.15 | 5.59 | 5.44 | 27.35 | 31.20 |
| Pseudo-L | 6.62 | 9.25 | 7.47 | 10.08 | 4.95 | 5.40 | 14.59 | 33.69 | 20.04 | 34.06 | 9.26 | 8.38 | 28.46 | 38.96 |
| LabelProp | 7.53 | 0.28 | 11.52 | 13.57 | 2.91 | 7.22 | 21.97 | 14.59 | 50.43 | 64.48 | 18.82 | 10.34 | 66.62 | 45.17 |
| OSL (ours) | 0.10 | **0.00** | **1.70** | **0.03** | **0.00** | **0.00** | **0.00** | 7.87 | **1.93** | **12.85** | **1.05** | **0.82** | **21.40** | **25.05** |
| Full-L | 0.23 | 0.00 | 1.02 | 0.00 | 1.19 | 0.86 | 7.44 | 8.46 | 1.17 | 9.45 | 7.84 | 0.84 | 22.08 | 26.29 |
| Full-T | 0 | 0 | 1.20 | 0.00 | 0 | 0 | 0 | 7.14 | 1.90 | 11.04 | 0 | 0 | 21.11 | 25.08 |

**Open-ended classification.** Table 1 shows the results of OSL and the baselines on open-ended classification tasks. On all tasks, OSL outperforms all baselines on both the subjective error and the model error. It is also worth noting that compared to the oracle Full-L with full label annotations, OSL still induces smaller subjective error, showing a strong capability of domain-level cognition that resembles the "ground truth" of human experts (Full-T). In addition, we visualize the features output by OSL on *Fashion Product Images*, as shown by Figure 4. The visualization shows that different models of OSL automatically focus on different label subspaces corresponding to different domains, which further complements our empirical results.

## 5 RELATED WORK

Apart from the formulation in this paper, open-ended data may also be formulated using the framework of multi-label learning with partial labels (Zhang & Zhou, 2014) or probabilistic density estimation, such as probabilistic concepts (Kearns & Schapire, 1994; Devroye et al., 1996) and the energy-based learning framework (LeCun et al., 2006). A fundamental difference between these formulations and OSL is that these methods learn a *unified* model, while our method explicitly encourages the learner to perform high-level data allocation and result in a set of independent models. A recent work by Su et al. (2020) studies a similar problem as ours where the goal of the learner is to learn from multi-task samples without task annotation. However, our work employs a different objective function and our method has formal theoretical justification.

Extensive literature has investigated the collaboration of multiple models (or modules) in completing one or more tasks (Doya et al., 2002; Shazeer et al., 2017; Meyerson & Miikkulainen, 2019; Alet et al., 2018; Nagabandi et al., 2019; Chen et al., 2020; Yang et al., 2020). The difference between our work and these works is that our multi-model architecture is driven by the inherent conflict in open-ended data without manual alignment between models and tasks, and we only allow a single low-level model to be invoked during a particular sampling episode.

## 6 DISCUSSION

An important open problem in machine learning is to move from highly specialized AI systems to agents with more general abilities. While various viewpoints exist, the notion of open-ended environments has been advocated by an increasing number of works (Goertzel & Pennachin, 2007; Clune, 2019; Silver et al., 2021). In this work, we attempt to formalize this notion in the context of supervised learning, and provide a general learning framework with theoretical guarantee. We hope that our work can facilitate future research in this direction.

**Limitations and future work.** As mentioned in Section 1, our formulation is limited to the fully-informative data where absolute predictions can be made given the inputs. While this assumption is valid in a variety of applications, it is interesting to develop methods that can also handle data with stochasticity. Other future work includes developing *lifelong learning* agents that benefit from the growing diversity of open-ended data as the sampling process continuously proceeds, and extending our framework to other learning regimes, e.g., semi-supervised learning and reinforcement learning.

ETHICS STATEMENT

This paper presents a new supervised learning framework which considers learning from the data directly sampled from open-ended environments such as the real world. Hence, our work may facilitate the development of more generally applicable artificial agents. Also, as mentioned in Section 1, our work has potential practical applications in federated learning and algorithmic fairness domains. Therefore, our work may promote the development of privacy-preserving and fair machine learning algorithms.

REPRODUCIBILITY STATEMENT

For all theoretical results in this paper (Proposition 1, Theorem 1 and Theorem 2), we provide complete proofs in the appendix (see Appendix C). For the open-ended regression and classification experiments, we provide PyTorch implementations in the supplementary material; the code for the classification experiment on *CIFAR-100* will be released if the paper is accepted since it requires additional pre-trained network backbones to run. The details of datasets, baselines and hyperparameters are provided in the appendix (see Appendix D).

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

## A DISCUSSION ON THE SUBJECTIVE FUNCTION

In Section 2.3, we derive our design of the subjective function using a EM-based maximum likelihood formulation. However, the exact equivalence between the E-step of hard EM and the subjective function has not been established, since it relies on the exact form of the loss function $\ell(y, y')$ and the conditional likelihood $p(y|x, h_k), k \in [K]$. As a complement, in the sequel we provide two examples where under the uniform prior, the exact equivalence between calculating the posterior $P(h = h_k|x, y)$ and the expected subjective function 5b can be obtained (hence their empirical counterparts are also equivalent). Recall that under the uniform prior, the E-step of hard EM yields $h = \arg\max_{h_k \in H} \log p(y|x, h_k)$.

**Example 1** (Regression with isotropic Gaussian joint likelihood)**.** Consider a regression task where we assume that given the random variable $(x, y)$, the joint distribution of its conditional likelihoods conforms to an isotropic Gaussian $\mathcal{N}(\boldsymbol{\mu}_K, \epsilon \boldsymbol{I}_K)$ where $\epsilon \in \mathbb{R}_+$ is a variance parameter and $\boldsymbol{I}_K$ is a $K \times K$ identity matrix. From $p(y|x, h_k) \propto \exp\left[-\frac{(y - \mu_k(x))^2}{2\epsilon^2}\right]$ and $\mu_k(x) = h_k(x)$ we have that $\arg\max_{h_k \in H} p(y|x, h_k) \Leftrightarrow \arg\min_{h_k \in H}\left[(y - h_k(x))^2\right]$. This equals to the expected subjective function with squared loss $\ell(y, y') = (y - y')^2$.

**Example 2** (Classification with independent categorical likelihoods)**.** Consider a multi-class classification task where we assume that given the random variable $(x, y)$, all conditional likelihoods conform to independent categorical distributions with parameters $(\boldsymbol{\lambda}_1, \boldsymbol{\lambda}_2, \cdots, \boldsymbol{\lambda}_K)$, where $\boldsymbol{\lambda}_k = (\lambda_{k1}, \cdots, \lambda_{kL}), \sum_{j=1}^{L} \lambda_{kj} = 1, k \in [K]$ and $L$ is the total number of classes. From $p(y|x, h_k) = \prod_{j=1}^{L} \lambda_{kj}(x)^{y_j}$, where $y = (y_1, \cdots, y_L) \in \{0, 1\}^L, \sum_{j=1}^{L} y_j = 1$ and $\boldsymbol{\lambda}_k(x) = h_k(x)$ we have that $\arg\max_{h_k \in H} \log p(y|x, h_k) \Leftrightarrow \arg\min_{h_k \in H}[-\sum_{j=1}^{L} y_j \log h_{kj}(x)]$, where $h_{kj}(x)$ denotes the predicted probability of the $j$-th class and $y_j$ is the corresponding ground truth. This equals to the expected subjective function with cross-entropy loss $\ell(y, y') = -\sum_{i=1}^{L} y'_i \log y_i$.

## B ALGORITHM PSEUDO-CODE

We provide the pseudo-code of OSL in Algorithm 1.

## C PROOFS OF THEORETICAL RESULTS

In this section, we provide the proofs of our theoretical results. For better exposition, we restate each theorem before its proof.

### C.1 PROOF OF PROPOSITION 1

**Proposition 1** (Form of the optimal solution)**.** *Assume that the target functions in all domains are realizable. Then, the following two propositions are equivalent:*

*(1) For all domain distributions $Q$ and data distributions $P_1, P_2, \cdots, P_N$, $er(H) = 0$.*

*(2) For each $d = \langle P, c \rangle$ in $D$, there exists $h \in H$ such that $\mathbb{E}_{x \sim P}\ell[h(x), c(x)] = 0$.*

*Proof.* The derivation of proposition (1) from proposition (2) is obvious. On the other hand, if proposition (2) is false, i.e., there exists $k \in [N]$ such that for all $j \in [K], h_j \neq c_k$. Then, we have

$$er(H) = \mathbb{E}_{c_i \sim Q} \min_{h \in H} er_{P_i}(h, c_i)$$
$$\geq q(c_k) \min_{h \in H} er_{P_k}(h, c_k)$$

From the above we know that $c_k \notin H$, thus there exists $P_k$ such that $er_{P_k}(h, c_k) > 0$ for every $h \in H$. This indicates that $er(H) > 0$, which is in contradiction with proposition (a). Therefore, proposition (2) must hold if proposition (1) is true. □

---

**Algorithm 1** OSL: Open-ended Supervised Learning

---

**Require:** Hypothesis set $H = \{h_1, h_2, \cdots, h_K\}$, sampling hyperparameters $m$ and $n$.
1: **for** $h = 1, 2, \cdots, T$ epochs **do**
2:     **for** $i = 1, 2, \cdots, m$ episodes **do**
3:         Sample data $Z^i = \{(x^{i1}, y^{i1}), (x^{i2}, y^{i2}), \cdots, (x^{in}, y^{in})\}$.
4:         Select a hypothesis $\hat{h}^i$ from the hypothesis set using the empirical subjective function 5a.
5:         Train the hypothesis $\hat{h}^i$ by minimizing the empirical episodic error 6.
6:     **end for**
7: **end for**

---

### C.2 PROOF OF THEOREM 1

We first revisit the definition of PAC learnability.

**Definition 2** (PAC learnability). *A target function set class $\mathbb{C}$ is said to be PAC learnable if there exists an algorithm $\mathcal{A}$ such that for any $\epsilon > 0$ and $\delta \in (0, 1]$, for all distributions $Q$ and distribution set $\mathcal{P}$, the following holds:*

$$P\left[er(H) \leq \epsilon\right] \geq 1 - \delta. \tag{10}$$

*A target function set class $\mathbb{C}$ is said to be PAC learnable if there exists an algorithm and a polynomial function $\mathrm{poly}(\cdot, \cdot, \cdot, \cdot)$ such that for any $\epsilon > 0$ and $\delta > 0$, for all distributions $Q$ and distribution set $\mathcal{P}$, the following holds for any sample size $mn \geq \mathrm{poly}(1/\epsilon, 1/\delta, \mathrm{size}(c))$ :*

The above definition can be viewed as an extension of the single-task PAC learnability (Haussler, 1990) that considers the problem of learning a single target function. Based on this definition, in the following we give the proof.

**Theorem 1.** *A necessary condition of the PAC learnability of OSL is $K \geq R$.*

*Proof.* According to Definition 2, if OSL is PAC learnable, there must exist an algorithm that outputs a hypothesis set $H$ with zero error, i.e., $er(H) = 0$ for every $Q$ and $\mathcal{P}$ (otherwise the inequality 10 will not hold for a small enough $\epsilon$ and $\delta < 1$). Proposition 1 indicates that this is equivalent to $c_i \in H, i \in [N]$, which is impossible if $K < R$ according to Definition 1. □

### C.3 PROOF OF THEOREM 2

We first introduce several technical lemmas.

**Lemma 1.** *Let $\{E_i\}_{i=1}^n$ be a set of events satisfying $P(E_i) \geq 1 - \delta_i$, with some $\delta_i \geq 0, i = 1, \cdots, n$. Then, $P\left(\bigcap_{i=1}^n E_i\right) \geq 1 - \sum_{i=1}^n \delta_i$.*

**Lemma 2** (Single-task generalization error bound (Vapnik, 2013)). *Let $A \leq \mathcal{Q}(z, \alpha) \leq B, \alpha \in \Lambda$ be a measurable and bounded real-valued function set, of which the Vapnik-Chervonenkis (VC) dimension (Vapnik & Chervonenkis, 1971) is $\mathsf{VC}(\mathcal{Q})$. Let $\{z_1, z_2, \cdots, z_n\}_{i=1}^n$ be data samples sampled i.i.d. from a distribution $P$ with size $n$. Then, for any $\delta \in (0, 1]$, the following inequality holds with probability at least $1 - \delta$:*

$$\left| \mathbb{E}_{z \sim P} \mathcal{Q}(z, \alpha) - \frac{1}{n} \sum_{i=1}^n \mathcal{Q}(z_i, \alpha) \right| \leq (B - A)\sqrt{\epsilon(n)} + \frac{1}{n}, \tag{11}$$

*where*

$$\epsilon(n) = \frac{\mathsf{VC}(\mathcal{Q})\left(\ln{2n}/\mathsf{VC}(\mathcal{Q}) + 1\right) - \ln{\delta/4}}{n}. \tag{12}$$

Then, we upper bound the error induced by the estimation of the expected subjective function 5b in each sampling episode of OSL, which is critical in bounding the generalization error. Recall that we

use superscripts to denote the sampling index, e.g., $d^i = \langle P^i, c^i \rangle$ denotes the $i$-th domain sample, which can be any domains in the domain set $D$. We define two shorthands as follows:

$$h_i^* := \arg\min_{h \in H} \mathbb{E}_{x \sim P^i} \ell\left[h(x), c^i(x)\right], i \in [m], \tag{13a}$$

$$\hat{h}_i^* := \arg\min_{h \in H} \sum_{j=1}^{n} \ell\left[h\left(x^{ij}\right), y^{ij}\right], i \in [m], \tag{13b}$$

where $i$ denotes the $i$-th sampling episode of OSL, $H$ is the hypothesis set.

**Lemma 3** (Subjective estimation error bound). *Let $\left\{\left(x^{i1}, y^{i1}\right), \cdots, \left(x^{in}, y^{in}\right)\right\}$ be episodic samples in the $i$-th $(i \in [m])$ sampling episode i.i.d. drawn from domain $d^i$ with size $n$. Then, for any $\delta \in (0,1]$, the following inequality holds uniformly for all hypothesis $h \in \mathcal{H}$ with probability at least $1 - \delta$:*

$$\frac{1}{m}\sum_{i=1}^{m}\frac{1}{n}\sum_{j=1}^{n} \ell\left[h_i^*\left(x^{ij}\right), y^{ij}\right] - \frac{1}{m}\sum_{i=1}^{m}\frac{1}{n}\sum_{j=1}^{n} \ell\left[\hat{h}_i^*\left(x^{ij}\right), y^{ij}\right]$$
$$\leq 2\sqrt{\frac{\mathsf{VC}(\mathcal{S})(\ln 2n/\mathsf{vc}(\mathcal{S}) + 1) - \ln \delta/8m}{n}} + \frac{2}{n}, \tag{14}$$

*where $\mathcal{S} := \{(x,y) \mapsto \ell[h(x;\theta), y]\}, \theta \in \Theta$ is the function set of the sample-wise error.*

*Proof.* We have the following decomposition:

$$\frac{1}{m}\sum_{i=1}^{m}\frac{1}{n}\sum_{j=1}^{n} \ell\left[h_i^*\left(x^{ij}\right), y^{ij}\right] - \frac{1}{m}\sum_{i=1}^{m}\frac{1}{n}\sum_{j=1}^{n} \ell\left[\hat{h}_i^*\left(x^{ij}\right), y^{ij}\right]$$

$$= \frac{1}{m}\sum_{i=1}^{m}\left\{\frac{1}{n}\sum_{j=1}^{n} \ell\left[h_i^*\left(x^{ij}\right), y^{ij}\right] - \mathbb{E}_{x \sim P^i} \ell\left[h_i^*(x), c^i(x)\right]\right\}$$

$$+ \frac{1}{m}\sum_{i=1}^{m}\left\{\mathbb{E}_{x \sim P^i} \ell\left[h_i^*(x), c^i(x)\right] - \mathbb{E}_{x \sim P^i} \ell\left[\hat{h}_i^*(x), c^i(x)\right]\right\}$$

$$+ \frac{1}{m}\sum_{i=1}^{m}\left\{\mathbb{E}_{x \sim P^i} \ell\left[\hat{h}_i^*(x), c^i(x)\right] - \frac{1}{n}\sum_{j=1}^{n} \ell\left[\hat{h}_i^*\left(x^{ij}\right), y^{ij}\right]\right\},$$

in which the original difference is decomposed into three terms. By definition 13a and 13b the middle term is non-positive. Using Lemma 2 by substituting $\mathcal{Q}$ with $\mathcal{S} = \{(x,y) \mapsto \ell[h(x;\theta), y]\}$ and replacing $\delta$ with $\delta/2m$, the first term and the last term can both be bounded by $\sqrt{\frac{\mathsf{VC}(\mathcal{S}) \ln (2n/\mathsf{vc}(\mathcal{S}) + 1) - \ln \delta/8m}{n}} + \frac{1}{n}$ with probability at least $1 - \delta/2$ (Lemma 1). Combining these two bounds using Lemma 1 completes the proof. $\square$

Now we can give the proof of the main theorem.

**Theorem 2** (Generalization error bound of OSL). *For any $\delta \in (0,1]$, the following inequality holds uniformly for all hypothesis sets $H \in \mathcal{H}^K$ with probability at least $1 - \delta$:*

$$er(H) \leq \widehat{er}(H) + \sqrt{\frac{\mathsf{VC}(\bar{\mathcal{S}}) \left(\ln 2m/\mathsf{vc}(\bar{\mathcal{S}}) + 1\right) - \ln \delta/12}{m}} + \frac{1}{m} \tag{15a}$$

$$+ \sum_{k=1}^{N}\left(\frac{m_k}{m}\sqrt{\frac{\mathsf{VC}(\mathcal{S})\left(\ln 2m_k n/\mathsf{vc}(\mathcal{S}) + 1\right) - \ln \delta/12N}{m_k n}} + \frac{1}{mn}\right) \tag{15b}$$

$$+ 2\sqrt{\frac{\mathsf{VC}(\mathcal{S})\left(\ln 2n/\mathsf{vc}(\mathcal{S}) + 1\right) - \ln \delta/24m}{n}} + \frac{2}{n}, \tag{15c}$$

*where $\bar{\mathcal{S}} := \{\langle P, c \rangle \mapsto \mathbb{E}_{x \sim P} \ell[h(x;\theta), c(x)]\}, \theta \in \Theta$ is the function set of the domain-wise expected error, $\mathcal{S} := \{(x,y) \mapsto \ell[h(x;\theta), y]\}, \theta \in \Theta$ is the function set of the sample-wise error, $m_k :=$*

$\sum_{i=1}^{m} \mathbb{1}\left(c^i = c_k\right)$ *is the sampling count of the target function from the k-th domain* $d_k(k \in [N])$, *and* $\mathsf{VC}(\cdot)$ *the Vapnik-Chervonenkis (VC) dimension (Vapnik & Chervonenkis, 1971).*

*Proof.* Combining the objectives 2 3 with the subjective function 5a 5b, we have the following decomposition:

$$er(H) - \widehat{er}(H) = \left\{ \mathbb{E}_{d_i \sim Q} \min_{h \in H} \mathbb{E}_{x \sim P_i} \ell\left[h(x), c_i(x)\right] - \frac{1}{m} \sum_{i=1}^{m} \min_{h \in H} \mathbb{E}_{x \sim P^i} \ell\left[h(x), c^i(x)\right] \right\}$$

$$+ \left\{ \frac{1}{m} \sum_{i=1}^{m} \min_{h \in H} \mathbb{E}_{x \sim P^i} \ell\left[h(x), c^i(x)\right] - \frac{1}{m} \sum_{i=1}^{m} \frac{1}{n} \sum_{j=1}^{n} \ell\left[h_i^*\left(x^{ij}\right), y^{ij}\right] \right\}$$

$$+ \left\{ \frac{1}{m} \sum_{i=1}^{m} \frac{1}{n} \sum_{j=1}^{n} \ell\left[h_i^*\left(x^{ij}\right), y^{ij}\right] - \frac{1}{m} \sum_{i=1}^{m} \min_{h \in H} \frac{1}{n} \sum_{j=1}^{n} \ell\left[h\left(x^{ij}\right), y^{ij}\right] \right\}.$$

By definition 13a and 13b we rewrite the equation above:

$$er(H) - \widehat{er}(H) = \left\{ \mathbb{E}_{d_i \sim Q} \mathbb{E}_{x \sim P_i} \ell\left[h_i^*(x), c_i(x)\right] - \frac{1}{m} \sum_{i=1}^{m} \mathbb{E}_{x \sim P^i} \ell\left[h_i^*(x), c^i(x)\right] \right\}$$

$$+ \left\{ \frac{1}{m} \sum_{i=1}^{m} \mathbb{E}_{x \sim P^i} \ell\left[h_i^*(x), c^i(x)\right] - \frac{1}{m} \sum_{i=1}^{m} \frac{1}{n} \sum_{j=1}^{n} \ell\left[h_i^*\left(x^{ij}\right), y^{ij}\right] \right\}$$

$$+ \left\{ \frac{1}{m} \sum_{i=1}^{m} \frac{1}{n} \sum_{j=1}^{n} \ell\left[h_i^*\left(x^{ij}\right), y^{ij}\right] - \frac{1}{m} \sum_{i=1}^{m} \frac{1}{n} \sum_{j=1}^{n} \ell\left[\hat{h}_i^*\left(x^{ij}\right), y^{ij}\right] \right\},$$

in which the generalization error of OSL is decomposed into three terms. By substituting $\mathcal{Q}$ in Lemma 2 by $\bar{\mathcal{S}} = \{\langle P, c\rangle \mapsto \mathbb{E}_{x \sim P} \ell\left[h(x;\theta), c(x)\right]\}$ and replacing $\delta$ with $\delta/3$, the first term can be bounded by $\sqrt{\frac{\mathsf{VC}(\bar{\mathcal{S}}) \ln\left(2m/\mathsf{vc}(\bar{s}) + 1\right) - \ln \delta/12}{m}} + \frac{1}{m}$ with probability at least $1 - \delta/3$. By replacing $\delta$ with $\delta/3$ in Lemma 3 we bound the last term by $2\sqrt{\frac{\mathsf{VC}(\mathcal{S})(\ln 2n/\mathsf{vc}(\mathcal{S}) + 1) - \ln \delta/24m}{n}} + \frac{2}{n}$ with probability at least $1 - \delta/3$. There remains the middle term for which we have

$$\frac{1}{m} \sum_{i=1}^{m} \mathbb{E}_{x \sim P^i} \ell\left[h_i^*(x), c^i(x)\right] - \frac{1}{m} \sum_{i=1}^{m} \frac{1}{n} \sum_{j=1}^{n} \ell\left[h_i^*\left(x^{ij}\right), y^{ij}\right]$$

$$= \frac{1}{m} \sum_{i=1}^{m} \left\{ \mathbb{E}_{x \sim P^i} \ell\left[h_i^*\left(x\right), c^i(x)\right] - \frac{1}{n} \sum_{j=1}^{n} \ell\left[h_i^*\left(x^{ij}\right), y^{ij}\right] \right\}$$

$$= \frac{1}{m} \sum_{k=1}^{N} \left\{ m_k \mathbb{E}_{x \sim P_k} \ell\left[h_k^*(x), c_k(x)\right] - \frac{1}{n} \sum_{j=1}^{nm_k} \ell\left[h_k^*\left(x_{kj}, y_{kj}\right)\right] \right\}$$

$$= \frac{1}{m} \sum_{k=1}^{N} m_k \left\{ \mathbb{E}_{x \sim P_k} \ell\left[h_k^*(x), c_k(x)\right] - \frac{1}{nm_k} \sum_{j=1}^{nm_k} \ell\left[h_k^*\left(x_{kj}, y_{kj}\right)\right] \right\}.$$

Recall that $m_k := \sum_{i=1}^{m} \mathbb{1}\left(c^i = c_k\right)$. In the above we re-arrange the total $m$ data batches according to the domains they belong to. With a little abuse of notation, in the third and fourth rows we use $h_k^*$ to denote the hypothesis that yields the smallest expected error in the $k$-th *domain* in $D$, i.e., $h_k^* = \arg\min_{h \in H} \mathbb{E}_{x \sim P_k} \ell\left[h(x), c_k(x)\right], k \in [N]$. Note that this is different from the definition 13a, where $h_i^*$ is defined upon the $i$-th *domain sample*. The above transformation aggregates the domain samples so that data samples from the same domains that emerge multiple times can be accumulated and jointly considered, which leads to a tighter and more realistic error bound. By Lemma 2, for every $k \in [N]$ and $\delta \in (0, 1]$, each inside term $\mathbb{E}_{x \sim P_k} \ell\left[h_k^*(x), c_k(x)\right] -$

$\frac{1}{nm_k}\sum_{j=1}^{nm_k}\ell\left[h_k^*\left(x_{kj},y_{kj}\right)\right]$ can be bounded by $\sqrt{\frac{\mathsf{VC}(\mathcal{S})\left(\ln 2m_k n/\mathsf{vc}(\mathcal{S})+1\right)-\ln \delta/12N}{m_k n}}+\frac{1}{m_k n}$ with probability at least $1-\delta$. By replacing $\delta$ with $\delta/3N$ we bound the whole second term by $\sum_{k=1}^{N}\left(\frac{m_k}{m}\sqrt{\frac{\mathsf{VC}(\mathcal{S})\left(\ln 2m_k n/\mathsf{vc}(\mathcal{S})+1\right)-\ln \delta/12N}{m_k n}}+\frac{1}{mn}\right)$ with probability at least $1-\delta/3$ (Lemma 1).

Finally, combining the above bounds for all three terms using Lemma 1 gives the result. $\qquad\square$

## D    EXPERIMENT DETAILS

In this section, additional details on the setup and settings of the experiment are provided. All experiments were conducted based on PyTorch (Paszke et al., 2019) using a NVIDIA 2080ti GPU. All data we used does not contain personally identifiable information or offensive content, and can be obtained from public data sources.

### D.1    REGRESSION

We consider three heterogeneous mapping functions in absolute, sinusoidal and logarithmic function families as below:

$$y = 2\,|x| - 2,$$
$$y = 2\sin\left(3x + \frac{\pi}{2}\right),$$
$$y = \frac{3}{2}\log\left(-x + \frac{5}{2}\right) - 1,$$

where the range of $x$ is $[-2, 2]$ for all three functions, yielding the open-ended data with $R = 3$. The sample hyperparameters are $m = 250, n = 2$, i.e., a total number of 500 data points are collected in 250 episodes, each with 2 samples in the data batch, and their underlying generation functions are randomly chosen from the three mapping functions.

**Network and optimizer.** A simple network architecture with 5 fully-connected linear layers, each with 32 hidden units, is adopted and trained with SGD with step size $\alpha = 0.05$, $momentum = 0.9$, and $\ell_2\ weight\ decay = 10^{-4}$.

### D.2    CLASSIFICATION

*Colored MNIST* is an extended version of the well-known character recognition dataset MNIST, in which each gray-scale digital images are randomly colored with 8 different colors. The dataset contains two parallel tasks of color and number classification ($R = 2$) with a total number of 60000 colored digits. The sample hyperparameters are $m = 60000, n = 1$.

*Fashion Product Images* (Aggarwal, 2019) is a dataset for automatic attribute completion and Q&A of clothing product images with multiple category labels in different domains. We choose 3 main parallel tasks (color, gender, category) with 8 main labels from the original dataset ($R = 3$), with 15000 images in total. The sampling hyperparameters are $m = 15000, n = 1$.

*CIFAR-100* (Krizhevsky & Hinton, 2009) is a classical benchmark for general image classification. It has a hierarchical structure with 20 superclasses and 100 classes, with 60000 images in total. Each superclass is from an upper-level "coarse" task, consisting of 5 "fine" classes, e.g., "insects" includes "bee, beetle, butterfly, caterpillar, cockroach". In our experiments, these two different kinds of labels are randomly provided given an image ($R = 2$). The sampling hyperparameters are $m = 30000, n = 2$.

**Network and optimizer.** In *Colored MNIST*, each network consists of 3 convolutional layers and 1 fully-connected layer. In *Fashion Product Images*, each network consists of 5 convolutional layers and 3 fully-connected layers. We use Adam optimizer (Kingma & Ba, 2015) with step size $\alpha = 0.002$ and $betas = (0.5, 0.999)$ for these two datasets. In *CIFAR-100*, for OSL and all baselines, we use a pre-trained DenseNet (Huang et al., 2017) backbone of DenseNet-L190-k40 for

feature extraction to ensure a fair comparison, and add 2 fully-connected layers after the DenseNet backbone. We use SGD as the optimizer with step size $\alpha = 0.1$ and $momentum = 0.9$.

### D.3 BASELINE DETAILS

In this section, we provide more details on the baselines.

#### D.3.1 REGRESSION

**MAML.** For MAML, we adapt a standard PyTorch implememtation from GitHub[1] with the same network with ours, and use the following hyperparameters:

$Shot = 2, Evalation = 100, Outer\ step\ size = 0.05, Inner\ step\ size = 0.015,$
$Inner\ grad\ steps = 2, Eval\ grad\ steps = 5, Eval\ iters = 10, Iterations = 20000.$

**Modular.** For modular meta-learning, we use the official PyTorch implementation from GitHub[2], and use the following hyperparameters:

$Shot = 2, Support = 2, Network = Linear\ 1 - 16 - 16 - 1, Num\_modules = 5,$
$Composer = Sum, meta\_lr = 0.003, Steps = 3000.$

Note that for Modular Meta-Learning a larger episodic sample number $n = 4$ (consisting of 2 support samples and 2 query samples) is adopted. Nevertheless, OSL still outperforms this approach using a smaller episodic sample number ($n = 2$).

#### D.3.2 CLASSIFICATION

**Probablistic concepts.** From the perspective of probability modeling, the relation between input $x$ and output $y$ is subject to a probability distribution $p(y \mid x)$, which is the learning target. In open-ended classification, when different domains share similar frequency, such a distribution can be approximated with the total probability formula:

$$p(y \mid x) = \sum_{h \in H} p(y \mid x, h) \cdot p(h) \approx \frac{1}{l_d} \sum_{h \in H} p(y \mid x, h)$$

For classification problems, in each domain $d$, the corresponding $p(y \mid x, h)$ is unimodal. Thus, their sum $p(y \mid x)$ is a multi-modal distribution, and a network trained with cross-entropy loss is still an unbiased estimation for it. The final prediction is the labels with top-$N$ conditional probabilities $p(y \mid x)$.

**Semi-supervised multi-label learning.** From the perspective of semi-supervised multi-label learning, the open-ended classification problem can be modeled as "multi-label learning with missing labels": consider the fully labeled data $x \to \boldsymbol{y} = \{y_i\}_{i=1}^N$, then, open-ended data provides only one label $y \in Y$ for each $x$, i.e., all other labels are missing. Therefore, existing semi-supervised learning approaches may be modified to handle such problems. We consider two representative techniques, including *Pseudo-label* and *Label propagation*. Both implementations are slightly modified to fit our tasks.

*Pseudo-label* randomly allocates additional "pseudo" labels to each input to compensate the missing labels. The learning machine is trained on the augmented dataset and then reevaluate the confidence of all pseudo labels according to its predictions, and all pseudo labels will iteratively be modified during training until convergence.

*Label propagation* builds a graph over the samples, where each node on the graph represents a data sample, and each edge represents the distance of two nodes it collects in the feature space. The labels are propagated on adjacent nodes until all samples are fully labeled. The feature space is also iteratively adjusted along during training.

**Full labels & full tasks.** Both these oracle baselines utilize manual annotations to tranform the open-ended data with $R > 1$ to conventional supervised data with $R = 1$. More concretely, for *Full*

---

[1] https://github.com/dragen1860/MAML-Pytorch (MIT license)
[2] https://github.com/FerranAlet/modular-metalearning (MIT license)

Table 2: Results of OSL on multi-dimensional regression task on gas sensor array under dynamic gas mixtures dataset. We report RMSE on each domain and the macro-average RMSE over both domains.

| Methods | RMSE (domain 1) | RMSE (domain 2) | RMSE (macro-average) |
|---|---|---|---|
| Vanilla (single model) | 76.5 | 89.7 | 83.1 |
| OSL (ours) | **34.0** | **72.6** | **53.3** |
| Oracle | 31.9 | 66.8 | 49.4 |

*labels*, label annotations from all domains are provided simultaneously as a "multi-hot" label vector for each input sample, resulting in a standard multi-label learning problem. For *Full tasks*, raw data is separated into single-class classification tasks according to additional manual task annotation, resulting in a standard multi-task learning problem.

### D.3.3    MEASURING THE SUBJECTIVE ERROR OF BASELINES IN CLASSIFICATION

Since the baselines in classification experiments do not explicitly assign a separate model to each domain, the subjective error metric 8 does not directly apply to these baselines. Hence, we estimate their subjective errors using another method: we directly select top-$N$ predictions of these methods and compare them with the label spaces of different domains. We define the "coverage" of the selected top-$N$ labels over domain $d$ as

$$\text{COVERAGE}(d) = \frac{1}{l_d} \sum_{i=1}^{l_d} \mathbb{1}(\exists \text{ top-}N \text{ labels for input } x_i \text{ that is in the label space of } d),$$

where $l_d$ denotes the total number of samples in domain $d$. The subjective error on $d$ is then calculated as $1 - \text{COVERAGE}(d)$.

## E    ADDITIONAL EXPERIMENTAL RESULTS AND VISUALIZATIONS

In this section, we provide additional experimental results and visualizations.

### E.1    MULTI-DIMENSIONAL OPEN-ENDED REGRESSION

To further demonstrate the efficacy of OSL in regression problems, we conducted experiments on a real-world multidimensional regression dataset from UCI machine learning repository: Gas sensor array under dynamic gas mixtures dataset (Fonollosa et al., 2015). This dataset contains the recordings of 16 chemical sensors exposed to two dynamic gas mixtures and the aim is to predict the concentrations of gases, with 417,8504 instances and 16-dimensional attributes. We treat each gas mixture as one domain, respectively representing Ethylene & Methane (domain 1) and Ethylene & CO (domain 2) gas mixtures, and randomly split both domains into training (90%) and test sets (10%).

In this task, we compare OSL (two models, trained on the union of both domains) with a vanilla single model regressor (trained on the union of both domains) and an oracle regressor with two models separately trained on each domain. We report root mean square error (RMSE) on each domain and the macro-average RMSE over both domains in Table 2. The results indicate that OSL benefits from its automatic data allocation process, surpassing the vanilla baseline that trains a single global model by a large margin and performs similarly with the oracle.

### E.2    SIMULATED CONCEPT SHIFT CLASSIFICATION

To further demonstrate the applicability of OSL, we conducted an experiment on *Fashion Product Images* with simulated concept shift between domains with the *same* label spaces. Concretely, we

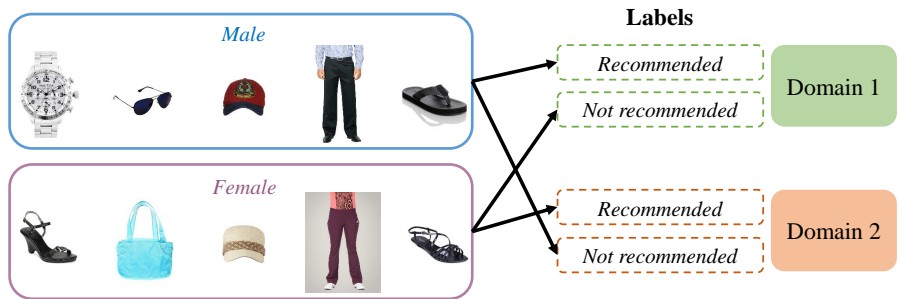

Figure 5: Open-ended classification tasks with simulated concept shift based on the *Fashion Product Images* dataset. Two domains indicate different recommendations based on the gender attribute.

Table 3: Results of OSL on simulated concept shift classification task. We report subjective and model errors on both domains.

| Methods | SUBERR | | MODERR | |
|---|---|---|---|---|
| | *Domain 1* | *Domain 2* | *Domain 1* | *Domain 2* |
| Vanilla (single model) | - | - | 49.5 | 50.5 |
| OSL (ours) | **5.37** | **5.37** | **7.40** | **9.65** |
| Full-T (oracle) | 0 | 0 | 7.14 | 6.53 |

randomly split the dataset into two domains according to the "gender" attribute: 50% of samples labeled as "Male" and 50% samples labeled as "Female" are assigned to domain 1 with "Male" relabeled as "1" and "Female" relabeled as "0", and other samples are assigned to domain 2 with "Male" relabeled as "0" and "Female" relabeled as "1". As shown in Figure 5, this setting resembles a simple scenario of concept shift caused by human preference: the label "1" can be interpreted as an indicator of "interested" or "recommended", while the label "0" can be interpreted as an indicator of "not interested" or "not recommended". Since different people may have different preferences (in this simulated experiment this is caused by gender – domain 1 and domain 2 respectively represents appropriate recommendations to male and female users), the data samples from different domains may possess different input-label conditional probability $P(y \mid x)$, albeit with the same label space.

Since the label space is binary and completely shared by both domains, multi-label baselines are inapplicable in this setting since they will always produce both labels "0" and "1" for every input and do not discriminate between different domains, which is unmeaningful. Therefore, we compare OSL with the single-model baseline (vanilla) and an oracle (Full-T) which separately train two models on both domains explicitly with data-domain correspondences known in advance. We report the model error and the subjective error of OSL in Table 3. Results show that the vanilla single-model baseline cannot learn meaningful prediction results since it lacks the mechanism to distinguish conflicting samples from different domains, while OSL achieves relatively small subjective errors and similar model errors as the Full-T oracle, which demonstrate that OSL is effective even in the context where different domains possess exactly the same label spaces yet different input-label relations.

### E.3  TRAINING AND INFERENCE TIME ON FASHION PRODUCT IMAGES

In Table 4, we compare the computational cost of OSL and other baselines in terms of training and inference time on the *Fashion Product Images* dataset. We measure the required wall-clock time (in seconds) for each method to reach convergence during training as well as the averaged wall-clock time for each method to predict all labels of one given input (in milliseconds). Concretely, for OSL and all baselines except for two oracles (Full-L and Full-T), we train the models for 50 epochs with about 15,000 images in every epoch; for Full-L and Full-T, we train for 10 epochs since these methods generally converge faster. For each method, we test its total inference time on the same 3,000 test samples randomly sampled from the test set and report the mean inference time on each test sample. All results are obtained with PyTorch using a NVIDIA 2080ti GPU.

Table 4: Wall-clock training and inference time of OSL and baselines on the *Fashion Product Images* dataset. The training time of ProbCon, Pseodu-L, LabelProp and OSL is measured over 50 epochs, and the training time of Full-L and Full-T is measured over 10 epochs. The inference time of all methods is measured using an average over 3,000 test samples.

| Methods | Training time (in seconds) | Inference time (in milliseconds) |
|---|---|---|
| ProbCon | 415 | 0.67 |
| Pseudo-L | 833 | 2.00 |
| LabelProp | 915 | 0.59 |
| OSL (ours) | 580 | 0.79 |
| Full-L | 159 | 0.68 |
| Full-T | 93 | 0.76 |

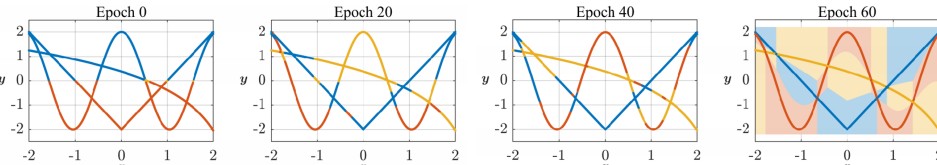

Figure 6: An example of the iteration process and the final decision boundaries of the subjective function in the regression task.

As shown in the table, the time cost of OSL is generally on par with or lower than baselines that also involve iterative training (Pseudo-L and LabelProp). Although the ProbCon baseline trains faster, it learns a global model without the mechanism of data allocation and thus performs significantly worse than OSL as we have shown. Meanwhile, compared with the Full-T oracle that knows the data-domain correspondences in advance, OSL only exhibits a little additional computational overhead. This indicates that although OSL incorporates an extra data allocation process implemented by the subjective function, this process only induces a limited computational cost since it only requires additional network forward process without loss backpropagation, which is in general efficient.

### E.4 ITERATION PROCESS ON THE REGRESSION TASK

In OSL, since the performance of the low-level models will impact the data allocation process of the high-level subjective function, the training of the subjective function and low-level networks can be regarded as an iterative process, as shown by Figure 6 with an example in open-ended regression, in which the different colored lines are designated to different subjects. All networks are randomly initialized, and in each iteration, each sample may be reallocated by the subjective function and used to further train the low-level networks. With the increasing of iterations, both subjective and model errors will reduce and converge along with the global loss. The last subfigure displays the final decision boundary of subjective function.

### E.5 FEATURE VISUALIZATION ON THE CLASSIFICATION TASK

The OSL approach can extract different semantics from the same input sample, and map them to different feature spaces. Figure 7 displays the features output under all subjectives, where each color represents a subjective and each point corresponds to an image in the dataset.

## F MORE DISCUSSION ON RELATED WORK

In this section, we provide more discussion on related work.

**Ensemble learning.** Ensemble learning approaches typically employ multiple models to cooperatively solve a given task (Dietterich, 2002; Zhang & Ma, 2012; Sagi & Rokach, 2018; Zhou, 2021). The prediction of each model is combined by weighting (boosting), majority voting (bagging) or

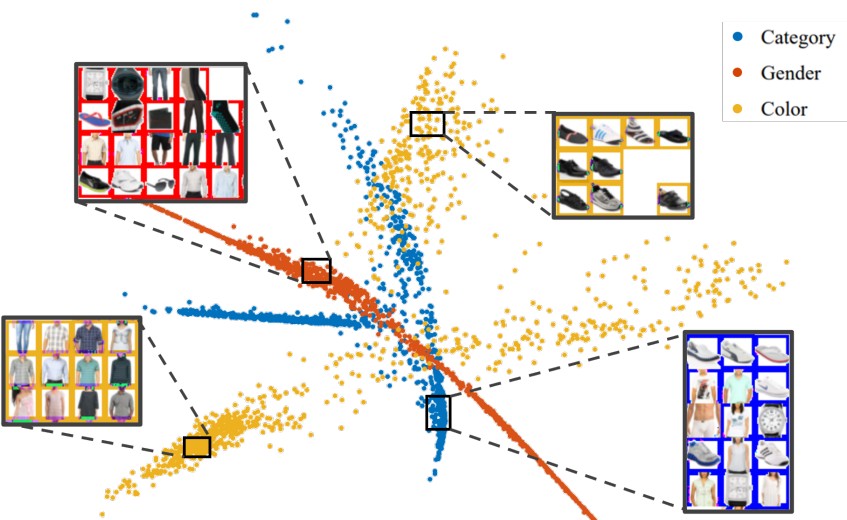

Figure 7: Additional visualization of image features by OSL on *Fashion Product Images*.

learning a second-level meta-learner (stacking). Since different models process the same set of data (although sometimes with different sample weights), in ensemble learning there is typically no explicit "hard" allocation process between the data and models. In contrast, the multi-model architecture of OSL is driven by the inherent conflict in open-ended data, and each model only handles a proportion of the whole dataset without overlapping.

**Domain adaptation and domain generalization.** Domain adaptation (Ben-David et al., 2010; Pan & Yang, 2010; Tan et al., 2018; Wang & Deng, 2018; Hoffman et al., 2018) and domain generalization (Blanchard et al., 2011; Muandet et al., 2013; Zhou et al., 2021) consider the scenario where the learner trained on one or multiple source domain(s) is transferred to one or multiple new target domain(s). Typically, domain adaptation focuses on the problem where there exist some labeled or unlabeled instances in the new domain, while domain generalization considers the setting where there the information of the new domains is inaccessible during training (i.e., zero-shot generalization). In other words, these formulations focus on the *adaptation* or *generalization* capability of the model on target domain(s). In contrast, OSL focuses on the multi-domain *training* process and considers the scenario where directly training a global model using the data from multiple conflicting domains is problematic, and aims to resolve this training issue by performing automatic data allocation. Another important difference is that in domain adaptation and domain generalization the data-domain correspondences are available, while OSL weakens this assumption by only assuming that the data from each sampling episode is obtained from the same domain.

