# OpenReview forum: "Subjective Learning for Open-Ended Data"
_ICLR.cc/2022/Conference — ICLR 2022 Submitted_

### Official Review · Reviewer_WErS · 2021-10-20

**Correctness:** 4
**Technical Novelty And Significance:** 3
**Empirical Novelty And Significance:** 3
**Recommendation:** 5
**Confidence:** 3

**Main Review:**

I like the general idea of the paper and appreciate the framework that I consider as novel and well suited for publication in general. I also find the technical part correct but did not check everything in detail. Concerns are caused by the baselines in the empirical comparison. The proposition of the framework clearly needs empirical support as the main argument is that a single classifier fails to learn all concepts properly. There need to be more information provided on the baselines to better understand whether they are appropriate or not. I am under the impression that the baselines may not be suitable to support the claims but this may be part of the rebuttal.

I would appreciate a better differentiation to other ensemble learning approaches as well as learning with many classes/labels as well as transfer learning approaches in the discussion.

**Summary Of The Paper:**

The authors present a novel framework for learning different target concepts. They argue that so far, a single classifier is supposed to learn mixed concepts but usually fail. Hence, they propose a framework for sampling instances according to only a single target concept and solving this problem with classical ERM and repeating for other concepts. They provide theoretical analyses and empirical insights.

**Summary Of The Review:**

Conceptually and theoretically strong but line of argument needs empirical evidence to work out. Though evidence is provided in form of empirical results on three data sets, the baselines remain unclear. This problem need to be solved before publication would be OK (could be straight forward to do so)

---

> ### Author Response · Authors · 2021-11-12
> **Response to reviewer WErS (1/2)**
>
> Thanks for your comments! We address your concerns in the following, please tell us if you have any other questions or concerns.
>
> ### Empirical support of the main argument
>
> We believe that the the main argument of our paper ("a single classifier fails to learn all concepts properly") is corroborated by our experiments: in the regression experiment the "vanilla" baseline is a single model that regress from multiple target functions (concepts) simultaneously, which trivially outputs their mean and fails to recover any of them; in the classification experiments, the "ProbCon" baseline represents a single classifier trained on data from multiple concepts (so that given an input, its prediction tends to be a probablistic "mixture" of different concepts, e.g., 50% "red" and 50% "1" in _ColoredMNIST_ ), yet it performs substantially worse than OSL. We believe that these empirical results suggest that learning a single model is problematic on conflicting data with multiple concepts.
>
> ### Details of the baselines
>
> Recall that the goal of OSL is to extract independent concepts from open-ended data where different concepts are mixed together. Therefore, we tried to select baselines that are most suitable in this setting with as few modifications as possible. In the sequel, we provide the details of baselines and the reason for choosing them in the regression and classification experiments respectively. Note that we provide more details on the implementation of these baselines in Appendix D.3 of the paper.
>
> + **Regression:** As far as we are concerned, there is no off-the-shelf baseline that can automatically disentangle the data from different domains in the open-ended regression setting. The most relevant approach in our view is meta-learning that learns a global model which can be fastly fine-tuned to fit each domain (this is different from OSL, which assigns different models to conflicting domains explicitly). Therefore, in the experiments, we compare OSL with two meta-learning approaches: MAML [1] and modular meta-learning [2] (as well as the "vanilla" baseline with a single model, as mentioned above), where MAML maintains a single tunable model and modular meta-learning maintains multiple tunable modules with their outputs summed together. We note that strictly speaking, meta learning approaches and OSL are designed for different problem settings (meta-learning approaches are designed for multi-task transfer across  similar domains and aim at fast adaptation, while OSL is designed for disentangling the data from different domains during training, and these domains may not be similar), hence it is not surprising that OSL outperforms meta-learning approaches in our setting.
>
> + **Classification:** In classification, the most relevant problem setting to OSL is multi-label learning [3], which considers the scenario where an input $x$ is related with a label set $\boldsymbol{y}=(y_1,y_2,\cdots,y_N)$. This setting is similar to OSL in terms that both OSL and multi-label learning considers the scenario where the same input may relate to multiple labels. However, the key difference is that multi-label learning requires that all labels in the label set are provided **simaltaneously**, while in OSL each data sample only contains **one** label $y_i\in\boldsymbol{y}$. Therefore, the setting of OSL can be alternatively modeled as "multi-label learning with missing labels", i.e., in each sample $N-1$ labels in $\boldsymbol{y}$ are missing and only one label remains (note that this is a very extreme setting). We thus compare two typical methods in this area, including Pseudo label and label propagation:
>
>     + _Pseudo label_ [4]: this method first randomly allocates additional "pseudo" labels to each input to compensate the missing labels. The learning machine is then trained on the augmented dataset and then reevaluate the confidence of all pseudo labels according to its predictions, and all pseudo labels will be iteratively updated along with the training of the model until convergence.
>
>     + _Label propagation_ [5]: this method builds a graph over the samples, where each node on the graph represents a data sample, and each edge represents the distance of two nodes it collects in the feature space. The labels are propagated on adjacent nodes until all samples are fully labeled. The feature space is also iteratively adjusted during training.
>
>     Also, as the ablation study we compare OSL with the single model baseline (ProbCon) and two oracles that respectively provide the learner with the full label set (Full-L) and true data-domain correspondences (Full-T).

---

> > ### Author Response · Authors · 2021-11-12
> > **Response to reviewer WErS (2/2)**
> >
> > ### Comparison with other works
> >
> > + **Ensemble learning:** Ensemble learning approaches typically employ multiple models to cooperatively solve a given task, where there is no allocation process between the data and models. In contrast, the multi-model architecture of OSL is driven by the inherent conflict in open-ended data, and each model only handles a proportion of the whole dataset without overlapping.
> >
> > + **Multi-label learning:** As we have discussed above, multi-label learning assigns each input with a label set containing multiple labels, but requires that all labels in the label set are provided simultaneously (which also yields a mapping rank $R=1$, but has a larger label space than conventional classification with unique ground-truth labels).
> >
> > + **Transfer learning:** Transfer learning considers the scenario where a classifier trained on one (or multiple) domains is transferred to a new domain (target domain). In other words, transfer learning focuses on the **adaptation** process on the target domain. In contrast, OSL focuses on the multi-domain **training** process and considers the scenario where directly training a global model using the data from multiple conflicting domains is problematic, and aims to resolve this training issue by performing automatic data allocation.
> >
> > ### References
> > [1] Chelsea Finn, Pieter Abbeel, and Sergey Levine. Model-agnostic meta-learning for fast adaptation of deep networks. In _ICML_, 2017.
> >
> > [2] Ferran Alet, Toms Lozano-Prez, and Leslie P. Kaelbling. Modular meta-learning. In _CoRL_, 2018.
> >
> > [3] Min-Ling Zhang and Zhi-Hua Zhou. A review on multi-label learning algorithms. _IEEE Transactions on Knowledge and Data Engineering_, 26(8):1819–1837, 2014.
> >
> > [4] Dong-Hyun Lee. The simple and efficient semi-supervised learning method for deep neural networks. In _ICML_, 2013.
> >
> > [5] Ahmet Iscen, Giorgos Tolias, Yannis Avrithis, and Ondrej Chum. Label propagation for deep semi-supervised learning. In _CVPR_, 2019.

---

### Official Review · Reviewer_E4u9 · 2021-10-31

**Correctness:** 3
**Technical Novelty And Significance:** 3
**Empirical Novelty And Significance:** 2
**Recommendation:** 6
**Confidence:** 2

**Main Review:**

**List strong and weak points of the paper. Be as comprehensive as possible.**

**Pros:**

* Generally, the paper is well written and easy to grasp the underlying ideas and concepts except how the data sampling works (see questions).
* The intorduction makes it very clear what the problem is this paper is trying to solve. (see additional feedback)
* A strength of Section 2+3 is that it bundles the theoretical formulas with intuitions in natural language, which makes it very clear at any time what the author trying to achieve or show.

**Cons:**

* While the generated regression example is great to gain intuitions on how OSL works and the influence of its hyperparameters, it's difficult to draw conclusions in regard to "real" regression problems. (see question)
* The proposed concept of the mapping rank seems to not account for label noise explicitly - but I would guess that in the presence of label noise there could be a rank > 1 even though there is only one sensible target function?
* The experimental evaluation appears to be rather specific and the baselines appear not directly related to other state of the art approaches. It would probably be a much stronger paper, if the authors showed that the proposed approach improves on some metric that is not specific to this submission
* The paper do not show training and prediction times for OSL and the other baselines.
* Minor/Typos:
  * in the introduction: would prefer to change "(image, label) pairs" -> "image-label pairs"
  * in the introduction: "can correlates" -> "can correlate" or "correlates"
  * last paragraph of the introduction: define PAC before use the abbreviation
  * first paragraph of Section 6: "facilicate" -> "facilitate"
  * page 4 last line: "objectives 2 3" -> "objectives 2 and 3"

**Clearly state your recommendation (accept or reject) with one or two key reasons for this choice.**

If the authors can clarify the points below, I vote for accept. The introduced approach OSL solves the problem of allocating data from different domains to their model. Currently, we often see very over-parametrized models that try to solve this implicitly. OSL could present a path to more efficient ML.

**Ask questions you would like answered by the authors to help you clarify your understanding of the paper and provide the additional evidence you need to be confident in your assessment.**

* Did you try at least a multidimensional generated regression? I think it would be interesting to see the method comparison in at least >1-D generated regression, better for a datasets, e.g., form OpenML.
* Could you clarify how the data sampling mechanism works?
  * It is still not clear to me whether the dataset X is sampled once or iteratively.

**Provide additional feedback with the aim to improve the paper. Make it clear that these points are here to help, and not necessarily part of your decision assessment.**

* The first paragraph after Definition 1 (introduction) is a bit lengthy. Shortening to its main points (difference between R=1 and R>1, metadata difficult to use for data allocation) could improve clarity
* Providing more description for Figure 2 and 3, especially, what the three lines show would add clarity
* You should consider to move the description of the DNNs from appendix to the paper.
* Providing an overview of training and prediction times of OSL and the baselines could be interesting for some readers.

**Summary Of The Paper:**

The paper introduces "Open-ended Supervised Learning (OSL)", a method that handles data form multiple domains (open-ended data). OSL leverages an Expectation-Maximization (EM) algorithm to train the "subjective function" that allocates models to data domains. The paper presents the theoretical foundations and empirical results in which it outperforms existing baselines for (generated) regression and classification tasks.

**Summary Of The Review:**

Interesting approach and novel concept of mapping rank, the experimental evaluation could provide more direct evidence for the value of the contribution.

Update after authors‘ response:
The authors addressed all points and provided convincing additional experiments, I’m hence increasing my score.

---

> ### Author Response · Authors · 2021-11-12
> **Response to reviewer E4u9**
>
> Thanks for your comments! We address your concerns in the following, please tell us if you have any other questions or concerns.
>
> ### Multidimensional regression
>
> We have conducted experiments on a real-world multidimensional regression dataset from UCI machine learning repository: Gas sensor array under dynamic gas mixtures dataset [1]. This dataset contains the recordings of 16 chemical sensors exposed to two dynamic gas mixtures and the aim is to predict the concentrations of gases, with 417,8504 instances and 16-dimensional attributes. We treat each gas mixture as one domain, respectively representing Ethylene & Methane (domain 1) and Ethylene & CO (domain 2) gas mixtures, and randomly split both domains into training (90%) and test sets (10%). We compare our method OSL with two models (trained on the union of both domains) with a vanilla single model  regressor (trained on the union of both domains) and an oracle regressor with two models separately trained on each domain. The results are listed in the following table, which indicates that OSL benefits from its automatic data allocation process, surpassing the vanilla baseline that trains a single global model by a large margin and performs similarly with the oracle.
>
> |Methods| RMSE (domain 1) $\downarrow$ | RMSE (domain 2) $\downarrow$ | RMSE (average) $\downarrow$ |
> |:---:|:---:|:---:|:---:|
> |Vanilla (single model)| 76.5 | 89.7 | 83.1 |
> |Oracle (two models)| 31.9 | 66.8 | 49.4 |
> |OSL (ours, two models)| 34.0 | 72.6 | 53.3 |
>
> ### Data sampling mechanism
>
> The dataset is sampled iteratively. Concretely, each domain has its own sub-dataset and the whole dataset is a union of these subsets. In each episode, a domain is randomly selected and a data batch with size $n$ is sampled from the corresponding subset. Different episode may sample data from different domains, i.e., different subsets of the whole dataset. Hence, the dataset (or any subsets of the dataset) may be sampled multiple times during training. We agree that the data sampling part in the paper is not clear enough and will revise this section accordingly for better exposition.
>
> ### Scenario with label noise
>
> Indeed, the concept of mapping rank is defined on the settings where there is no label noise. If there is noise on labels, then the mapping rank of data is inherently larger than one and can induces problems during data allocation. However, _we believe it may be fundamentally difficult to perform data allocation in the presence of label noise without additional priors_ : for example, if an input $x$ is mapped to label $y_1$ with probability $0.5$ and label $y_2$ with probability $0.5$, it seems impossible for one to distinguish whether this stochasticity is induced by different concepts/contexts or just "pure label noise" without other information. We leave more analyses on this point as future work.
>
> ### Training and inference time cost
>
> We did not accurately record the time cost of OSL and other baselines. However, as a brief overview, the training and the inference time cost of OSL are usually similar to the time cost of other baselines except for the vanilla single-model baseline (which usually trains faster but tends to fail in our setting). We will report detailed training and inference time costs in a future version of the paper.
>
> ### Minor/typos
>
> Thanks for additional feedback on improving the expression of the paper and correcting the typos, which we found very valuable and helpful. We will revise the paper accordingly according to your suggestions.
>
> ### Reference
> [1]  J. Fonollosa and R. Huerta. Gas sensor array under dynamic gas mixtures data set. https://archive.ics.uci.edu/ml/datasets/Gas+sensor+array+under+dynamic+gas+mixtures, 2015.

---

> ### Comment · Reviewer_E4u9 · 2021-12-04
> **Convincing additional experiments**
>
> Thanks a lot for the additional experiments!

---

### Official Review · Reviewer_tMyf · 2021-11-02

**Correctness:** 4
**Technical Novelty And Significance:** 3
**Empirical Novelty And Significance:** 3
**Recommendation:** 8
**Confidence:** 3

**Main Review:**

Strengths of the paper are:
-  The paper studies the important problem of learning on multiple domains with open-ended data. The proposed approach is novel, simple yet effective, as clearly demonstrated in both the theoretical and experimental results
- The paper is very well contextualised in the existing literature
- The paper is very well-written, organised and the claims are sufficiently justified

Weaknesses of the paper are:
- A few aspect are not entirely clear in the writing of the paper. For example, in page 3 it is mentioned that "In the above setting, an episodic sample number parameter n is introduced to maintain the local consistency of data, implicitly assuming that we are able to sample a size-n data batch at a time from each domain." - does this mean that at each time all domains should have corresponding data batches or that each domain is sampled sequentially one after the other? Please clarify
- There are some typos in the paper, e.g. Section 2.4 page 4

**Summary Of The Paper:**

This paper proposes a framework for open-ended data by learning a subjective functions that allows to represent multiple domains without interference. The framework works in two stages: 1) evaluating a set of candidate hypotheses for each domain using a batch of data for that domain, 2) training the hypothesis with the smallest error  for each domain. The paper also proposes two new evaluation metrics, one to determine the error of the subjective function and one for calculating the model error in domain prediction. The authors provide both theoretical and experimental analyses of the proposed framework. In terms of theoretical analyses, the authors provide some guarantees in terms of learnability and generalisation error. In terms of experimental results, the authors test both regression and classification tasks with data of multiple labels and hierarchical labels. Comparisons to other approaches such as ProbCon, Pseudo-L and LabelProp in Colored MNIST, CIFAR-100 and Fashion Product Images datasets demonstrate gains of the proposed approach versus counterparts.

**Summary Of The Review:**

I find this paper very interesting. The proposed approach is novel, yet simple, and clearly works well for the problem of learning from open-ended data. I would recommend to accept this paper.

---

> ### Author Response · Authors · 2021-11-12
> **Response to reviewer tMyf**
>
> Thanks for your comments and your approval for our work! We address your concerns in the following, please tell us if you have any other questions or concerns.
>
> ### Clarification on the sampling process
>
> We assume that the data in each episode is sampled from only one domain (but the learner do not know which domain it is sampled from); different episodes may correspond to different domains or the same domain determined by a uniform domain-level sampling. We do not require a strict sequential sampling over the domains.
>
> ### Typos
>
> Thanks for pointing out the typos, we will revise the paper and correct them accordingly.

---

### Official Review · Reviewer_R5W1 · 2021-11-03

**Correctness:** 3
**Technical Novelty And Significance:** 3
**Empirical Novelty And Significance:** 3
**Recommendation:** 6
**Confidence:** 3

**Main Review:**

While the intuition behind the problem and the proposed solution is
clear, the method is not clearly explained. In section 2.3, the
connection between the M and E steps is not clearly stated. More
importantly, the difference between conflicting and non-conflicting
domains is unclear. From the overall algorithm, it seems to me that
domains that do not share labels will be assigned to different
hypotheses (even if, in principle, they could be mutually-exclusive
labels), because a hypothesis previously trained on a different set of
labels will not predict the new labels. If this is the case, the only
non-conflicting domains are those that share the same label set but
maybe have differences in the input (like in a domain-adaptation
setting).  In this case the method seems an overkill, as one can
easily match the domains in the episodes by matching their labels (and
the subjective error should be minimized this way). The experiments
seems to suggest that this is indeed the setting, because different
domains correspond to different label sets.

Another unclear part is how the method is applied in testing. Assuming
that testing is made on an episode corresponding to a single domain,
how is/are the appropriate hypothesis/hypotheses selected? If the
predictions of all of them are collected, this seems to boil down to a
simple collection of multiclass classifiers (not sure what happens in
the regression case).

Depending on the specific setting (to be clarified), there could be
other (possibly simpler) competitors that are more appropriate. The
cross-domain fertilization is not evaluated in the experiments as far
as I can tell (apart from the matching of domains with the same
labels).

**Summary Of The Paper:**

The paper proposes a learning setting involving a sequence of domains
and a collection of hypotheses, During learning, each domain should be
assigned to the most appropriate hypothesis by iteratively grouping
together non-conflicting domains and keeping apart conflicting ones,
so as to minimize the number of hypotheses learned and leverage
cross-domain information when appropriate.

**Summary Of The Review:**

The paper is unclear in the description of the method, so much that
some trivial solutions seem even more appropriate for the setting.

Substantial clarifications are needed to properly evaluate the quality
of the contribution.

AFTER REBUTTAL:

The authors did manage to address many of my concerns in their detailed rebuttal and extended experimental evaluation. The difference between the proposed setting and the existing alternatives is now clearer and the advantage of the method better highlighted thanks to the novel experiments. Some aspects are still a bit unclear (e.g. how the method is applied at the test phase), but I am now more inclined towards acceptance.

---

> ### Author Response · Authors · 2021-11-12
> **Response to reviewer R5W1 (1/2)**
>
> Thanks for your comments! We are sorry that some details of our method have not been explained in a very clear manner, which led to some misunderstanding on several important points regarding our work. We hope that we can make these points more clear during the rebuttal and accordingly revise the paper to prevent the confusion for future readers. We address your concerns in the following, please tell us if you have any other questions or concerns.
>
> ### The difference between conflicting and non-conflicting domains
>
> As mentioned in the introduction, in our work the "conflict" between data samples or domains emerges when **the same input is mapped to different outputs**, corresponding to different **input-conditional label probabilities** $P(y|x)$ (**not marginal label probabilities** $P(y)$). Hence, _two domains may be **non-conflicting** even if they possess **different** label sets (e.g., MNIST and CIFAR10 are not conflicting since there is no overlapping image $x$); and two domains may be **conflicting** even if they possess the **same** label set (since the relation between inputs and labels may not be the same)_. This point is crucial, since it indicates that our method can be used in many scenarios where some "trivial solutions" raised in the review, e.g., label matching, is not applicable (see the next point for concrete examples).
>
> **OSL on conflicting and non-conflicting domains:** Our theory guarantees that when the number of hypotheses is sufficient, conflicting domains will be assigned to different hypotheses to minimize the global error (Theorem 1). However, non-conflicting domains may or may not be assigned to the same hypothesis depending on the number of hypotheses and the similarity of $P(y|x)$ among the domains: for example, consider three domains D1, D2 and D3 and two hypotheses H1 and H2, where D1 and D2 are conflicting and D3 is not conflicting with D1 or D2, then the data from D1 and D2 will be assigned to different hypotheses and the data from D3 will be assigned to either H1 or H2 even if it has completely different label sets from both D1 and D2; if we have three hypotheses instead, then the data from D3 will be assigned to H1/H2 if its $P(y|x)$ is similar to D1/D2 (to be more exact, as long as the model trained on D1 or D2 yields smaller loss on the data drawn from D3 compared to a randomly initialized new model) or H3 otherwise.
>
> ### On potential alternatives of OSL
>
> In your review you mentioned that "... one can easily match the domains in the episodes by matching their labels". Although this approach is effective when different domains have _strictly disjoint_ label sets, we argue that **this setting is very restrictive and the label matching approach is not applicable in at least the following two important settings, while OSL still works**:
>
> + **Regression:** in regression problems, all domains usually have the same label set $\mathbb{R}$ as in our open-ended regression experiments in Section 4.
>
> + **Classification with cross-domain concept shifts:** there are many scenarios where different domains have the **same** label set yet different input-label mappings (i.e., different $P(y|x)$): consider training a recommendation system where different people may have different preferences over a fixed set of goods. Similar scenarios are also common in federated learning where the data is distributed on multiple clients, and in algorithmic fairness research where the data is collected from different populations, as we have mentioned in the introduction of the paper.
>
> Also, we are not sure aboue the exact method(s) the "cross-domain fertilization" in the review refers to. We will appreciate if you can make this more clear for us to further compare it with OSL.

---

> > ### Author Response · Authors · 2021-11-12
> > **Response to reviewer R5W1 (2/2)**
> >
> > ### Testing process of OSL
> >
> > Concretely, the testing process of OSL can take two forms, which respectively correspond to the two evaluation metrics proposed in our paper (Section 4.2):
> >
> > + Given a domain $d$, the rate of inconsistent data allocations on all input-label pairs $(x,y)$ from $d$ is measured. This corresponds to the "**subjective error**" metric in the paper and measures the learner's ability to perform accurate data allocations.
> >
> > + For each input $x$, the predictions of all models are collected and compared with the ground-truth labels from all domains. This corresponds to the "**model error**" metric in the paper and measures the learner's ability to make accurate in-domain predictions. Importantly, _we do not think that this is trivial or "boil down to a simple collection of multiclass classifiers"_, as **independently extracting different concepts from conflicting data is exactly the aim of OSL, which cannot be achieved by directly training a collection of multiclass models on all domains**.
> >
> > Also, given a single domain with a small number of labeled samples and a large number of unlabeled samples, we can first select the hypothesis based on the labeled samples using the subjective function, and then use the selected hypothesis to predict the unlabeled ones (note that some labeled samples are needed to identify the domain). **This corresponds to the "single-domain" testing scenario mentioned in the review.** Because if both the subjective error and the model error is small, then our method should also yield good performance in this setting. Hence, we do not introduce additional metrics to measure the performance of this test process explicitly.
> >
> > ### More explanations on the connection between OSL and hard-EM
> >
> > As discussed in Section 2.3 in the paper, the iteration process of OSL can be connected with the hard-EM algorithm. Concretely, consider the $i$-th episode, the E-step of hard-EM seeks to find a hypothesis that maximize the posterior $P(h_k|Z^i)$ given a data batch $Z^i = \{ (x^{ij}, y^{ij}) \} \_{j=1}^n$, which is equivalent to the functionality of the subjective function (choosing a hypothesis for a given data batch) under certain conditions (see Appendix 1 for a detailed discussion); given $h = \arg\max_{h_k} P(h_k | Z^i)$, the M-step of hard-EM seeks to maximize the conditional log likelihood $\sum_{j=1}^n\log p(y^{ij} | x^{ij},h)$ under the hypothesis $h$, which equals to updating the selected hypothesis by minimizing the (empirical) prediction loss in the episode.

---

> > > ### Author Response · Authors · 2021-11-19
> > > **Additional concept shift classification experiment**
> > >
> > > To further complement our claim that OSL can also work well in the concept shift setting where different domains possess **the same label set** (besides regression), we have conducted an additional concept shift classification experiment based on the Fashion Product Images dataset used in the paper. Concretely, we randomly split the dataset into two domains according to the "gender" attribute: 50% of samples labeled as "Male" and 50% of samples labeled as "Female" are assigned to domain 1 with "Male" relabeled as "1" and "Female" relabeled as "0", and other samples are assigned to domain 2 with "Male" relabeled as "0" and "Female" relabeled as "1". This setting resembles a simple scenario of concept shift caused by human preference: the label "1" can be interpreted as an indicator of "interested" or "recommended", while the label "0" can be interpreted as an indicator of "not interested" or "not recommended". Since different people may have different preferences (in this simulated experiment this is caused by gender – domain 1 and domain 2 respectively represents appropriate recommendations to male and female users), _the data samples from different domains may possess different input-conditional label probabilities $P(y | x)$, albeit with the same label space and the same marginal label distribution $P(y)$_.
> > >
> > > We find that OSL is effective in this setting, just as in the settings where different domains have different label sets. We provide more details on this experiment and the exact results in the revised paper (**Appendix E.2, Figure 5, Table 3** on pages 20 and 21).

---

> ### Comment · Reviewer_R5W1 · 2021-11-30
> **after rebuttal**
>
> Thanks for your detailed answer and improvements to the paper. See my review for updates.

---

> > ### Author Response · Authors · 2021-12-01
> > **Thanks for the updates**
> >
> > Thank you for raising your score! We have found your comments very helpful in the revision, and will continue to revise our manuscript to improve its clarity.

---

### Author Response · Authors · 2021-11-19
**General response: Paper revision**

We thank the reviewers for their valuable comments and constructive suggestions. Accordingly, we have revised the paper and conducted additional experiments motivated by several comments. We summarize the major improvements of the paper during revision as follows:

**Concept shift classification experiment:** We conducted an additional concept shift classification experiment where different domains have the same label set yet different input-conditional label probabilities, which further demonstrate the applicability of OSL. (Appendix E.2, Table 3, Figure 5)

**Multi-dimensional regression experiment:** We conducted an additional multi-dimensional regression experiment based on a real-world regression dataset, demonstrating the effectiveness of OSL in real-world open-ended regression problems. (Appendix E.1, Table 2)

**More explanations on the sampling process and method:** We added more explanations on the sampling process and the connection between OSL and hard-EM, and modified several expressions to improve clarity. (Section 2.1, Section 2.3)

**More experimental and baseline details:** We added more discussion on baselines and added more descriptions to the captions of Figure 2 and Figure 3 to improve clarity. (Section 4.1, Appendix D.3)

**More discussion on related work:** We included more discussion on the related work on multi-label learning (Section 4.1), ensemble learning, domain adaptation and domain generalization (Appendix F).

**Training and inference time cost:** We reported training and inference time cost of OSL and baselines on the classification task on the Fashion Product Images dataset. (Appendix E.3, Table 4)

**Presentation improvement:** We modified or simplified some expressions in the introduction to improve clarity, and corrected typos in the paper.

All major updates have been highlighted using red text in the revised paper PDF.

---

### Author Response · Authors · 2021-12-04
**Looking forward to further comments**

Dear Reviewers and AC,

We sincerely appreciate your time and efforts in reviewing this paper. We have responded to the concerns raised by reviewers and revised our manuscript accordingly. Also, we would greatly appreciate if reviewers E4u9 and WErS could let us know if our response has addressed your questions and concerns.

Again, thank you for your time!

Best Regards,

Authors of Paper4677

---

### Decision · Program_Chairs · 2022-01-20

**Decision:**

Reject

**Comment:**

Two reviewers increased their scores after considering the responses from the authors, and all reviewers are somewhat positive. However, the increased scores are still 6 only, and as the area chair, I have concerns about the foundations of this research.

The authors write "there is no off-the-shelf baseline that can automatically disentangle the data from different domains in the open-ended regression setting." This is not true for the standard situation of a mixture of regression lines, as in Section 4.1. Completely standard EM (not necessarily hard EM) will solve this problem, as long as the individual lines (sinusoids etc.) can be represented easily by the EM components. Another standard method that should be another baseline is a mixture-of-experts neural network.

One thing that EM cannot handle is learning the number of components in a mixture model, as in learning the _k_ in _k_-means. To the extent that "open-ended" here refers to a new approach for this problem, with mathematical guarantees, it is interesting. But this point of view needs more explanation.

The paper begins "A hallmark of general intelligence is the ability of handling open-ended environments, which roughly means complex, diverse environments with no manual task specification." If there is one aspect of natural environments that is crucial and fundamental, it is the presence of noise. However, starting theoretically with Definition 1 and empirically with Section 4.1, the authors work in a world of deterministic functions. This mismatch undermines the conceptual significance of the paper.

Perhaps because of the universality of noise, the authors do not present a real-world dataset or task for which the OSL method is directly natural or applicable. Rather, they impose restricted specifications on datasets such as MNIST and introduce metrics that are novel, hence hardly natural, undermining the empirical significance of the paper.